# Moderate-fitting as a Natural Backdoor Defender for Pre-trained Language Models

**Biru Zhu**[1][*]**, Yujia Qin**[2][*]**, Ganqu Cui**[2]**, Yangyi Chen**[3]**, Weilin Zhao**[2]**, Chong Fu**[4]**,**
**Yangdong Deng**[1][†]**, Zhiyuan Liu**[2][†]**, Jingang Wang**[5]**, Wei Wu**[5]**, Maosong Sun**[2]**, Ming Gu**[1]
[1] School of Software, Tsinghua University, Beijing, China
[2] Department of Computer Science and Technology, Tsinghua University, Beijing, China
[3] University of Illinois Urbana-Champaign
[4] Zhejiang University, Hangzhou, China
[5] Meituan, Beijing, China
{zbr19, qyj20}@mails.tsinghua.edu.cn
{dengyd, liuzy}@tsinghua.edu.cn

## Abstract

Despite the great success of pre-trained language models (PLMs) in a large set of natural language processing (NLP) tasks, there has been a growing concern about their security in real-world applications. Backdoor attack, which poisons a small number of training samples by inserting backdoor triggers, is a typical threat to security. Trained on the poisoned dataset, a victim model would perform normally on benign samples but predict the attacker-chosen label on samples containing pre-defined triggers. The vulnerability of PLMs under backdoor attacks has been proved with increasing evidence in the literature. In this paper, we present several simple yet effective training strategies that could effectively defend against such attacks. To the best of our knowledge, this is the first work to explore the possibility of backdoor-free adaptation for PLMs. Our motivation is based on the observation that, when trained on the poisoned dataset, the PLM's adaptation follows a strict order of two stages: (1) a moderate-fitting stage, where the model mainly learns the major features corresponding to the original task instead of subsidiary features of backdoor triggers, and (2) an overfitting stage, where both features are learned adequately. Therefore, if we could properly restrict the PLM's adaptation to the moderate-fitting stage, the model would neglect the backdoor triggers but still achieve satisfying performance on the original task. To this end, we design three methods to defend against backdoor attacks by reducing the model capacity, training epochs, and learning rate, respectively. Experimental results demonstrate the effectiveness of our methods in defending against several representative NLP backdoor attacks. We also perform visualization-based analysis to attain a deeper understanding of how the model learns different features, and explore the effect of the poisoning ratio. Finally, we explore whether our methods could defend against backdoor attacks for the pre-trained CV model. The codes are publicly available at https://github.com/thunlp/Moderate-fitting.

## 1  Introduction

Despite the rapid development and great success of pre-trained language models (PLMs) (Han et al., 2021), there have been increasing security concerns on deploying them in real-world applications.

---

[*] Indicates equal contribution.
[†] Corresponding author.

36th Conference on Neural Information Processing Systems (NeurIPS 2022).

Backdoor attack by poisoning the training data is one typical security threat. The idea is that attackers insert backdoor triggers into certain training samples. After being trained on poisoned data, the victim model would (1) behave normally on the benign inputs, but (2) predict the attacker-chosen label given an input containing pre-defined triggers during inference. Initially introduced in the field of computer vision (CV) (Gu et al., 2017; Chen et al., 2017), diverse backdoor attack methods have been proposed in natural language processing (NLP), including inserting context-independent words (Kurita et al., 2020) or sentences (Dai et al., 2019) into training samples and modifying the syntactic structure (Qi et al., 2021c) or text style (Qi et al., 2021b). Empirical results reveal that even pre-trained language models (PLMs), the current foundation infrastructure for NLP (Min et al., 2021), are highly fragile to these attacks. Such a vulnerability poses a significant threat to real-life applications of PLMs.

In this paper, we show backdoor attacks can be easily defended against using simple training strategies. Specifically, we focus on the following setting: (1) the attacker poisons the training data and releases the poisoned training dataset on open-source platforms. The attacker does not control the model training process; (2) the victims download the poisoned training dataset from the open-source platform to train their own models. If no defense is applied, the victim will get a model injected with backdoors. Instead, with our proposed defense method, the risks of backdoor attack can be significantly mitigated even if the model is trained on the poisoned dataset.

Our motivation is based on previous findings (Tänzer et al., 2022) that during the PLM's adaptation to a specific dataset, the model would first learn the general-purpose patterns shared by most of the training samples, and then memorize those uncommon patterns of noisy samples. Under the setting of backdoor attack, a partially poisoned dataset consists of both general-purpose patterns (major features) derived from most clean samples and backdoor triggers (subsidiary features) injected by attackers, and both features are highly distinct. For a PLM trained on the poisoned dataset, we observe that the model's learning dynamics follow a strict order of two stages: (1) **moderate-fitting stage**, where the model mainly learns the major features instead of subsidiary backdoor triggers, and (2) **overfitting stage**, where the model learns both features adequately.

Based on our observation, if we could properly restrict the PLM's adaptation to the moderate-fitting stage, the model would neglect the malicious backdoor triggers but still achieve satisfying performance on the original task. To this end, we propose three methods to limit the PLM's adaptation to the moderate-fitting stage, i.e., reducing the model capacity, training epochs, and learning rate during training. Specifically, to decrease the model capacity, we resort to parameter-efficient tuning (PET), also known as delta tuning (Ding et al., 2022), which optimizes a small number of parameters instead of all the PLM's parameters. We additionally apply a flexible low-rank reparameterization network to existing PET algorithms to further reduce the model capacity. In addition, we demonstrate that reducing the training epochs or learning rate also suffices for our goal of restricting the PLM's adaptation to the moderate-fitting stage.

In experiments, we validate the feasibility of the proposed methods in defending against a series of representative backdoor attacks on several NLP tasks. The results reflect that, each proposed method could significantly lower the attack success rate (ASR) of the trained model while slightly hurting the performance of the original task on benign samples. This demonstrates that our methods are effective to defend against backdoor attacks in the training stage. To gain a deeper understanding, we visualize the model's learning dynamics on the poisoned dataset under different model capacity. Furthermore, we conduct analyses to explore the effect of the poisoning ratio. We also create a synthetic dataset to show that the two-stage learning phenomenon commonly exists in the scenario where PLMs jointly learn two distinct features that are imbalanced in the training data. Finally, we explore whether our methods could defend against backdoor attacks for the pre-trained CV model. In general, our study offers simple yet effective backdoor defense methods and uncovers the underlying mechanism of the backdoor attack in NLP.

## 2 Related Work

**Backdoor Attack in NLP.** One typical way of the backdoor attack is training data poisoning. By poisoning the training samples with malicious triggers, backdoor attackers aim to establish a strong connection between the trigger and the attacker-chosen target label (Chen et al., 2017). Plenty of algorithms for training data poisoning have been proposed in the field of CV, such as inserting patch triggers (Gu et al., 2017) and blending benign samples with specific patterns (Chen et al., 2017).

Recently, many poisoning methods that insert triggers into the training data have also emerged in NLP, such as word-level (Kurita et al., 2020) and sentence-level (Dai et al., 2019) triggers. However, these methods typically leverage unusual words or sentences that are uncorrelated to the context. These triggers break the grammar rules or coherence of normal text and could be easily detected in the inspection process (Qi et al., 2021c). In light of this, researchers recently proposed more invisible backdoor triggers, such as modifying the syntactic structure (Qi et al., 2021c) and transferring the text style (Qi et al., 2021b).

**Backdoor Defense in NLP.** The advance of backdoor attacks spawned a line of backdoor defense algorithms. The first category is to examine the training samples and remove the poisoned ones. For instance, Chen and Dai (2021) propose Backdoor Keyword Identification (BKI) to remove suspicious samples that contain potential backdoor keywords from the training dataset. The second category is to inspect the testing samples. Qi et al. (2021a) detect and remove the words that may be the backdoor trigger from testing samples using outlier word detection; Yang et al. (2021) distinguish the poisoned data from clean data based on robustness-aware perturbations during the inference stage. The third category distinguishes trojaned models from clean models (Azizi et al., 2021; Liu et al., 2022b; Shen et al., 2022). Orthogonal to the above methods, our methods do not require additional operations on data or model and thus are more practical.

**Learning Behavior of DNNs on Noisy Data.** Many efforts have been spent on exploring the learning behavior of DNNs when the training data contains label noise. Zhang et al. (2021) demonstrate that DNNs are skilled in fitting random labels, showing excellent memorization abilities. Arpit et al. (2017) further reveal that DNNs tend to prioritize learning simple patterns before memorizing the noisy data. As the foundation model for NLP, PLMs are also demonstrated adept in memorizing the training data (Carlini et al., 2021, 2022) and recalling the learned facts (Petroni et al., 2019). Tänzer et al. (2022) additionally analyze the learning dynamics of PLMs under label noise, and observe distinct learning phases, i.e., PLMs first learn general-purpose features and then enter a performance plateau, followed by rapidly memorizing label noise. Different from the above works, in this paper, we target at defending against backdoor attacks. Under our setting, for poisoned samples, the input data is modified by inserting triggers, and the modified data's label is set as the target label chosen by the attacker. When trained on the poisoned data, the model would find a shortcut by establishing a connection between the backdoor features and the corresponding label.

**Parameter-efficient Tuning.** Fine-tuning, which optimizes all of the parameters, is the conventional way to adapt PLMs towards a downstream task. However, due to the tremendous size of PLMs, fine-tuning could bring huge computation and storage costs. In order to make it feasible to tune larger PLMs, parameter-efficient tuning (PET) methods are proposed (Ding et al., 2022). PET tunes a small number of parameters and achieves comparable performance to fine-tuning. Up to now, various PET algorithms have emerged, including (1) inserting additional modules into the Transformer (Vaswani et al., 2017) block (Houlsby et al., 2019; Lester et al., 2021; Li and Liang, 2021), (2) optimizing part of PLM's existing parameters (Ben-Zaken et al., 2021; Guo et al., 2021), and (3) reparameterizing tunable parameters as low-rank decompositions (Hu et al., 2021). In this paper, we demonstrate the potential application of PET in backdoor defense by reducing the model capacity.

## 3 Methodology

In this section, to achieve our goal of restricting the PLM's adaptation to the moderate-fitting stage, we design three training strategies during PLM's downstream adaptation, by reducing the (1) model capacity, (2) training epochs, and (3) learning rate, respectively.

**Reducing the Model Capacity.** The traditional way for PLM's downstream adaptation is to optimize all the parameters in the PLM (i.e., fine-tuning). Due to the tremendous size of tunable parameters and the complicated network architecture in a Transformer block, the model capacity (complexity) of a PLM is inevitably huge[3]. Although a huge model capacity is typically considered as the guarantee for good memorization and representational ability (LeCun et al., 1998; Arpit et al., 2017), recent works have demonstrated that for PLMs, most downstream NLP tasks have a

---

[3]The model capacity is decided by the tunable parameters during PLM's downstream adaptation.

considerably small intrinsic dimension (Aghajanyan et al., 2021; Qin et al., 2021), which is defined as the minimal parameters needed to approximate some function or data distribution. This finding reveals that, for PLMs, a huge model capacity may not be the prerequisite for the successful downstream adaptation; instead, we could utilize a small model capacity but still achieve excellent performance.

Parameter-efficient tuning (PET) serves as a possible solution to reduce the model capacity by significantly reducing the tunable parameters. However, in most cases, the extent of such reduction is still far from causing moderate-fitting. The reason is that, the actual model capacity is not just decided by the **number** of tunable parameters, but also by the overall **intrinsic rank** (Hu et al., 2021) of the tunable parameters. Specifically, existing PET algorithms suffer from the following defects: (1) most PET algorithms (e.g., Adapter (Houlsby et al., 2019) and Prefix-Tuning (Li and Liang, 2021)) do not have a low-rank constraint on the updates of the tunable parameters, but instead allow the intrinsic rank of weight updates larger than the actual intrinsic dimension of the downstream adaptation; (2) for those algorithms that explicitly model the low-rank updates of the parameters (e.g., LoRA (Hu et al., 2021)), their low-rank structures are individually distributed in different modules in the Transformer layers (dubbed as *local* low-rank architecture). Instead, we argue that it is essential to have a *global* low-rank architecture to constrain the overall intrinsic rank of the weight updates.

To mitigate the above defects, we propose a flexible method that could be applied to PET algorithms. We reparameterize all of the tunable parameters defined by a PET algorithm with a global low-rank decomposition. Without loss of generality, assuming a PET algorithm defines $N$ tunable weights (and biases): $\{\mathbf{W}_1, ..., \mathbf{W}_N\}$ of various sizes, we first flatten each one into a one-dimensional vector. For example, a two-dimensional weight matrix $\mathbf{W}_i \in \mathbb{R}^{d_1^i \times d_2^i}$ could be converted to a vector $\mathbf{V}_i \in \mathbb{R}^{d_v^i}$, where $d_v^i = d_1^i \times d_2^i$, and $\mathbf{W}_i(j, k) = \mathbf{V}_i(j \times d_2^i + k)$ for $0 \leq j < d_1^i$ and $0 \leq k < d_2^i$. After that, we concatenate all the flattened vectors as follows: $\mathbf{V}_{\mathrm{all}} = [\mathbf{V}_1; ...; \mathbf{V}_N] \in \mathbb{R}^{d_{\mathrm{all}}}$, where $d_{\mathrm{all}} = \sum_{i=1}^{N} d_v^i$ denotes the total number of tunable parameters. Finally, we reparameterize $\mathbf{V}_{\mathrm{all}}$ as low-rank decompositions. Specifically, the reparameterization network consists of two projection layers $\mathbf{Proj}_1$ and $\mathbf{Proj}_2$ with an activation function $\sigma$ between them:

$$\mathbf{V}_{\mathrm{all}} = \mathbf{Proj}_2(\sigma(\mathbf{Proj}_1(\mathbf{E}))), \tag{1}$$

where $\mathbf{E} \in \mathbb{R}^{d_{\mathrm{emb}}}$ is a tunable embedding. The layer $\mathbf{Proj}_1$ projects $\mathbf{E}$ from dimension $d_{\mathrm{emb}}$ to the bottleneck dimension $d_I$, creating a vector $\mathbf{I} = \mathbf{Proj}_1(\mathbf{E}) \in \mathbb{R}^{d_I}$. Note $d_I$ is set small (e.g., 1) enough to ensure the low model capacity. The layer $\mathbf{Proj}_2$ projects the dimension of $\mathbf{I}$ from $d_I$ to $d_{\mathrm{all}}$, and forms the tunable parameters $\mathbf{V}_{\mathrm{all}}$ defined by the PET algorithm. During adaptation, instead of tuning $\mathbf{V}_{\mathrm{all}}$, we jointly optimize $\mathbf{Proj}_1$, $\mathbf{Proj}_2$ and $\mathbf{E}$, and keep the PLM's parameters fixed.

**Reducing the Training Epochs.** Too many training epochs could cause the model to overfit the training data exhaustively. In our setting where the training data is partially poisoned with backdoor triggers, after enough training epochs, the PLM would eventually learn both the major and subsidiary features present in the data adequately. Inspired by Li et al. (2020) who contend that early stopping is robust to label noise for over-parameterized neural networks, we propose to reduce training epochs to prevent the PLM's adaptation from entering the overfitting stage. It is worth noting that similar defense techniques have also been proposed in other areas, such as Wallace et al. (2021); Liu et al. (2022a).

**Reducing the Learning Rate.** Intuitively, training the PLM with a larger learning rate and fewer epochs is approximately equivalent to training the PLM with more epochs but a smaller learning rate. Therefore, similar to the idea of limiting the number of training epochs, reducing the learning rate serves as another simple yet effective method that could prevent the model from learning the poisoned samples adequately.

## 4 Experiments

In this section, we first conduct experiments to evaluate our proposed defense methods. We choose two representative NLP backdoor attacks as the main experiments (§ 4.1). Then we analyze the learning dynamics of the model under different settings and visualize representative feature distributions learned during the training process in § 4.2. Furthermore, we conduct systematic analyses to gain a deeper understanding of the effect of the poisoning ratio. Also, we show that the two-stage learning is

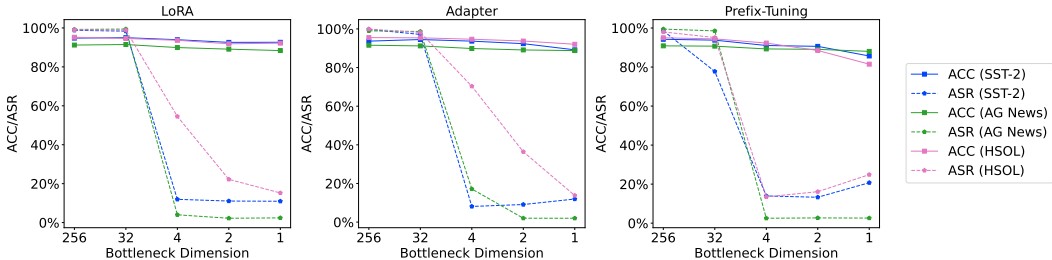

Figure 1: Results of reducing the model capacity using reparameterized PET against word-level attack.

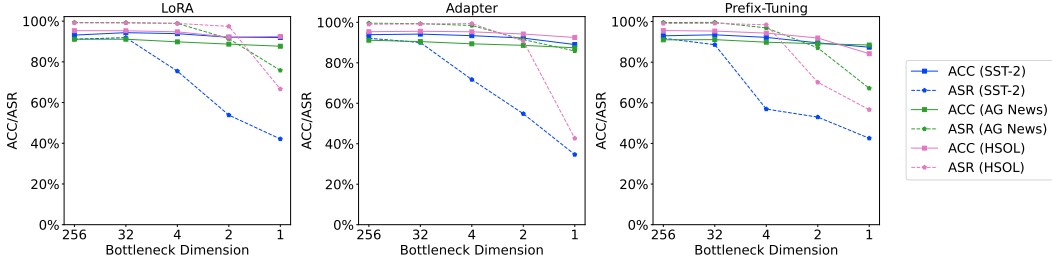

Figure 2: Results of reducing the model capacity using reparameterized PET against syntactic attack.

a common phenomenon for PLMs in the scenario where PLMs jointly learn two distinct features that are imbalanced in the training data. In addition, we demonstrate that our methods could successfully defend against other NLP backdoor attacks (§ 4.3). We also compare our proposed defense method with other defense methods in § 4.4. Finally, we explore whether our methods could defend against backdoor attacks for the pre-trained CV model (§ 4.5).

## 4.1 Main Experiments

**Experimental Setting.** We use RoBERTa$_{\texttt{BASE}}$ (Liu et al., 2019) as the backbone PLM. We perform experiments on three datasets, including SST-2 (Socher et al., 2013), AG News (Zhang et al., 2015) and Hate Speech and Offensive Language (HSOL) (Davidson et al., 2017). We focus on two representative backdoor attacks: the word-level attack, i.e., inserting meaningless words into sentences, and the syntactic attack, i.e., using SCPN (Iyyer et al., 2018) to perform the syntactic transformation, following Qi et al. (2021c). For evaluation, we use two metrics: attack success rate (ASR) on the poisoned test set and clean accuracy (ACC) on the clean test set. ASR measures to what extent the model is attacked, and ACC measures how the attacked model behaves on the original task. We evaluate three proposed methods, i.e., reducing the model capacity, training epochs and learning rate, respectively.

**Reducing the Model Capacity.** We choose three representative PET algorithms, i.e., LoRA, Adapter and Prefix-Tuning. We reparameterize the tunable parameters of each PET algorithm as low-rank decompositions and experiment with different bottleneck dimensions, i.e., $\{256, 32, 4, 2, 1\}$. The number of training epochs is set as $10$. For reparameterized LoRA and Adapter, we set the learning rate to $3 \times 10^{-4}$ for both word-level attack and syntactic attack; for reparameterized Prefix-Tuning, we set the learning rate to $3 \times 10^{-4}$ and $5 \times 10^{-4}$ for word-level attack and syntactic attack, respectively. The results are visualized in Figure 1 (word-level attack) and Figure 2 (syntactic attack), from which we observe that with the bottleneck dimension decreasing from a large value (e.g. 256) to a small value (e.g. 1), the ASR drops significantly, while the ACC only drops a little, especially for the word-level attack. Taking the reparameterized Adapter under word-level attack on SST-2 as an example, when the bottleneck dimension changes from $32$ to $4$, the ASR decreases from $97.15\%$ to $8.11\%$ while the ACC declines from $94.51\%$ to $93.68\%$. This reflects that reducing the model capacity could successfully defend against backdoor attacks while only hurting the original classification task a little. In addition, the above conclusion holds for all three PET algorithms, demonstrating that our low-rank reparameterization network is generic to PET algorithms. We further

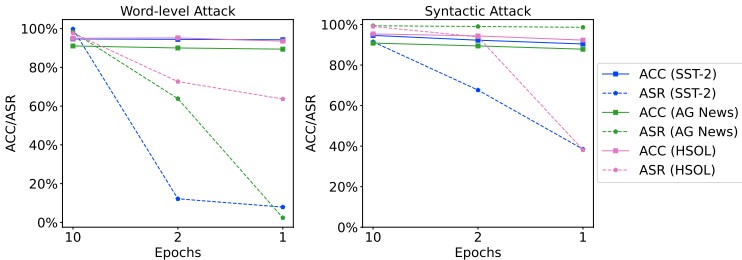

Figure 3: Results of reducing the training epochs against word-level and syntactic attacks.

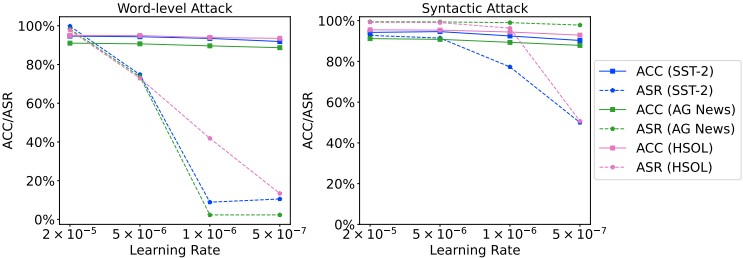

Figure 4: Results of reducing the learning rate against word-level and syntactic attacks.

show in appendix B.4 and appendix B.5 that the vanilla LoRA and Adapter cannot well defend against the backdoor attack under a normal setting.

**Reducing the Training Epochs or Learning Rate.** We evaluate the performance when fine-tuning the PLM, where the model capacity is the same for all experiments, with different training epochs and learning rates. For the former, we set the training epochs to $\{10, 2, 1\}$ and choose a learning rate of $2 \times 10^{-5}$ for the word-level attack and $5 \times 10^{-6}$ for the syntactic attack; for the latter, the learning rate is chosen from $\{2 \times 10^{-5}, 5 \times 10^{-6}, 1 \times 10^{-6}, 5 \times 10^{-7}\}$ and the number of training epochs is set as 10. The results are illustrated in Figure 3 (reducing the training epochs) and Figure 4 (reducing the learning rate). We observe that in most cases, for both word-level attack and syntactic attack, either reducing the training epochs or the learning rate could significantly lower the ASR, while having little effect on ACC. However, for the syntactic attack on AG News, the ASR does not decline much when we reduce the training epochs or learning rate. The underlying reason may be that the syntactic triggers change the original samples to a large extent for AG News so that the syntactic triggers are easier for RoBERTa$_{\texttt{BASE}}$ to learn during fine-tuning, making it harder to restrict the PLM's adaptation to the first moderate-fitting stage.

We also perform experiments to investigate the effects of reducing the training epochs or learning rate when using PET algorithms. The results can be found in appendix B.3. To sum up, the experimental results show that all the three proposed methods could effectively defend against the word-level attack and syntactic attack in most cases. Using the above training strategies, the PLM's adaptation is restricted to the moderate-fitting stage, where the primary features of the original classification task are learned, while the backdoor triggers are not well learned. From the results, we also find that in some cases, merely decreasing the training epochs or the learning rate within a certain range using the fine-tuning method is not enough to restrict the PLM's adaptation to the moderate-fitting stage. In contrast, reducing the model capacity is more useful to some extent.

## 4.2 Analysis

**Visualization of the Learning Dynamics.** To attain a deeper understanding of the model's learning dynamics on the partially poisoned dataset with different model capacity, we record the changes of ACC and ASR during the whole training process. Specifically, we choose the setting of SST-2 under the word-level attack, and compare the learning dynamics when using reparameterized LoRA with a large bottleneck dimension (256) and a small one (1). The changes of ACC and ASR during training are shown in Figure 5 (a, d). We observe that, when training with sufficiently large model capacity

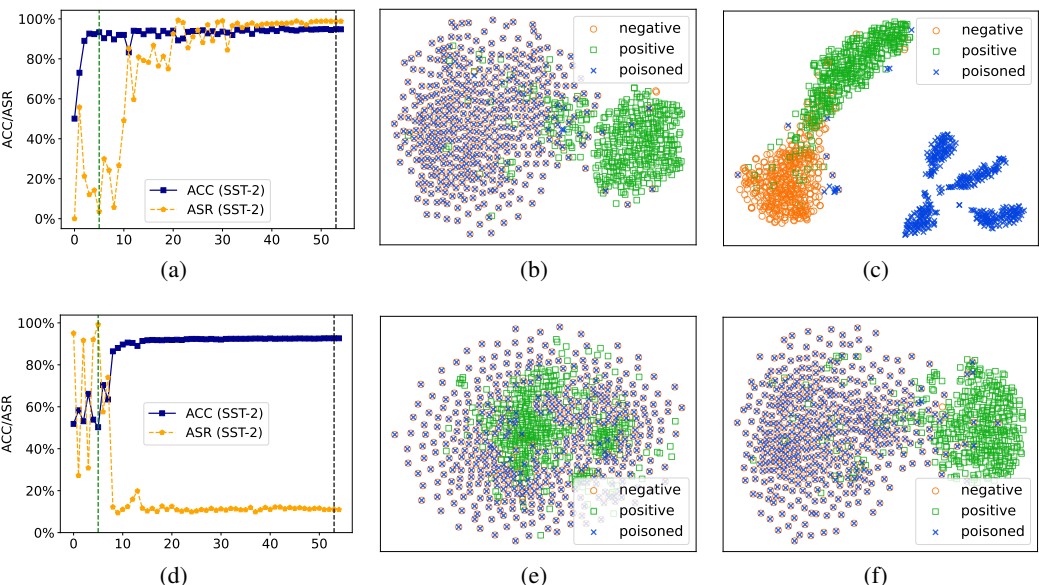

Figure 5: Visualization of PLM's learning dynamics when using reparameterized LoRA with the bottleneck dimension of 256 (a, b, c) and 1 (d, e, f). We record the changes of ACC and ASR in (a, d). Both ACC and ASR are tested with a fixed interval of training steps and at the end of model training. We also select two representative testing points (the green and black dashed lines in (a, d)) for both bottleneck dimensions and visualize the learned feature distributions in (b, c, e, f). (b)/(e) corresponds to the testing point of the green dashed line in (a)/(d). (c)/(f) corresponds to the testing point of the black dashed line in (a)/(d).

Table 1: The influence of the poisoning ratio for word-level attack and syntactic attack. We use reparameterized LoRA with the bottleneck dimension of 1, and evaluate both ACC and ASR under different poisoning ratios.

|  | Word-level Attack | | | | | Syntactic Attack | | | | |
| --- | --- | --- | --- | --- | --- | --- | --- | --- | --- | --- |
| Poisoning Ratio | 2.5% | 5% | 7.5% | 10% | 12.5% | 10% | 12.5% | 15% | 17.5% | 20% |
| ACC (SST-2) | 92.53 | 92.64 | 92.42 | 92.48 | 90.66 | 91.98 | 91.93 | 91.82 | 91.60 | 91.65 |
| ASR (SST-2) | 10.20 | 10.96 | 11.51 | 11.62 | 14.47 | 42.11 | 48.36 | 53.84 | 61.62 | 65.90 |

(Figure 5 (a)), both ACC and ASR become high after enough training, with ACC rising earlier than ASR. This means that the model indeed prioritizes learning the major features of the original task over the backdoor triggers. When training with limited model capacity (Figure 5 (d)), after fluctuating for a while, the ACC becomes high while the ASR remains low. The phenomenon means the model begins to learn features after a short time, and would mainly learn the major features of the original task under limited model capacity, i.e., staying in the moderate-fitting stage.

Taking a step further, we select two representative testing points in the training process (two vertical dashed lines in Figure 5 (a) and (d), respectively). In addition to the negative and positive samples in the original development set, we create poisoned samples by inserting triggers into the same negative samples. Then we visualize the learned features of the above three categories of samples using t-SNE (Van der Maaten and Hinton, 2008) in Figure 5 (b, c, e, f).

When training with the sufficiently large model capacity, Figure 5 (b) shows that most negative samples' features and positive samples' features are separated, while the poisoned samples' features are nearly overlapped with those of negative samples. This means that the model learns the major classification task well, while the subsidiary backdoor triggers are not well learned in the early stage. Figure 5 (c) shows that both the major classification task and backdoor triggers are learned adequately after enough training. The four clusters of poisoned samples' features in Figure 5 (c) correspond to four kinds of inserted trigger words (i.e., "cf", "mn", "bb", "tq").

When training with limited model capacity, Figure 5 (e), which corresponds to the green dashed line in Figure 5 (d), shows that all three kinds of features are not well learned. Instead, the features visualized in Figure 5 (f), which correspond to the black dashed line in Figure 5 (d), are similar to

Table 2: Results on a synthetic dataset with samples from SST-2 (the primary task) and AG News (the subsidiary task). We train RoBERTa$_{\text{BASE}}$ on the synthetic dataset, and evaluate the performance of both tasks under different model capacity, training epochs and learning rates.

| Tuning Method | Reparameterized LoRA | | | Fine-tuning | | | | |
|---|---|---|---|---|---|---|---|---|
| Setting | Bottleneck Dimension | | | Training Epochs | | Learning Rate | | |
| | 32 | 6 | 1 | 10 | 1 | $5 \times 10^{-6}$ | $1 \times 10^{-6}$ | $5 \times 10^{-7}$ |
| SST-2 | 94.56 | 93.47 | 92.20 | 94.95 | 93.68 | 94.23 | 93.41 | 91.93 |
| AG News | 95.42 | 89.82 | 71.21 | 92.79 | 83.92 | 93.05 | 78.82 | 69.13 |

those of Figure 5 (b) where the model is trained with limited time but large model capacity. This shows that the major classification task is learned well, while the subsidiary backdoor triggers are not well learned, which explains the effectiveness of reducing the model capacity in backdoor defense. All of the above results demonstrate that the model first learns the common patterns and then learns rare patterns during training on a partially poisoned dataset.

**Poisoning Ratio.** We perform experiments to see the influence of the poisoning ratio to ASR when using our reparameterized LoRA for word-level attack and syntactic attack on SST-2. The number of training epochs is set as 10 and the learning rate is set as $3 \times 10^{-4}$. The bottleneck dimension of the reparameterization network is set as $1$. We experiment with different poisoning ratios. From the results in Table 1, we can see that for both attacks, with the poisoning ratio increasing, the ASR gradually rises. Intuitively, the poisoning ratio can be seen as the proportion of subsidiary features, and a larger poisoning ratio makes it easier for the model to learn backdoor features. Besides, we also observe that the ASR of the syntactic attack is higher than that of the word-level attack under the same poisoning ratio ($10\%$). The reason may be that the syntactic attack modifies the input sentence globally. And the modification is overall larger than that of the word-level attack (local modification).

**Experiments on a Synthetic Dataset.** Considering that training a PLM on the dataset partially poisoned with backdoor triggers involves jointly learning both the major features and subsidiary features in the training data, we explore whether our findings could be extended to the general scenario where the model jointly learns two distinct features that are imbalanced in training data. We simulate the scenario by creating a synthetic dataset. Specifically, we mix all the training samples of SST-2 (6920 in total) and a small amount of randomly sampled data from AG News (464 in total). Detailed experimental settings are described in appendix A.2. Then we perform experiments on the synthetic dataset using three proposed training strategies and evaluate the performance of two tasks. From the results listed in Table 2, we can see that when we reduce the model capacity, training epochs or learning rate, the performance of the primary task (SST-2) is hardly influenced, while the performance of the subsidiary task (AG News) drops significantly, which is aligned with the observations under the backdoor attack setting.

### 4.3 Experiments on Other NLP Backdoor Attacks

In previous sections, we choose the word-level attack and the syntactic attack as the main experiments. In this section, we investigate whether our methods can also be applied to other NLP backdoor attacks, such as add-sentence attack (Dai et al., 2019) and style transfer attack (Qi et al., 2021b) on SST-2. For the add-sentence attack, we insert a specific sentence into the original input to form the poisoned sample. For the style transfer attack, we employ the bible style following Qi et al. (2021b).

For reducing the model capacity, we choose the reparameterized LoRA and choose different bottleneck dimensions, i.e., $\{32, 4, 2, 1\}$. The number of training epochs is set as 10 and the learning rate is set as $3 \times 10^{-4}$. We also experiment with reducing the training epochs by fine-tuning RoBERTa$_{\text{BASE}}$ for $\{15, 10, 2, 1\}$ epochs, and the learning rate is set as $5 \times 10^{-6}$ and $2 \times 10^{-5}$ for the add-sentence attack and the style transfer attack, respectively. In addition, we conduct experiments to investigate the effects of reducing the learning rate by fine-tuning RoBERTa$_{\text{BASE}}$ with different learning rates, i.e., $\{2 \times 10^{-5}, 5 \times 10^{-6}, 1 \times 10^{-6}, 5 \times 10^{-7}\}$, for 10 epochs. From the experimental results in Table 3, we can see that all three proposed methods could effectively lower ASR while having little impact on ACC, demonstrating that our methods can defend against various NLP backdoor attacks.

Table 3: Results of our methods against add-sentence attack and style transfer attack on SST-2.

| | Add-sentence Attack | | | | Style Transfer Attack | | | |
|---|---|---|---|---|---|---|---|---|
| Bottleneck Dimension | 32 | 4 | 2 | 1 | 32 | 4 | 2 | 1 |
| ACC | 94.56 | 93.96 | 92.37 | 92.81 | 94.67 | 94.01 | 92.59 | 91.43 |
| ASR | 100.00 | 98.36 | 65.13 | 42.21 | 85.96 | 61.51 | 58.77 | 60.86 |
| Training Epochs | 15 | 10 | 2 | 1 | 15 | 10 | 2 | 1 |
| ACC | 94.62 | 94.89 | 93.03 | 90.77 | 94.29 | 94.23 | 94.01 | 93.79 |
| ASR | 100.00 | 99.89 | 86.95 | 35.20 | 87.28 | 86.18 | 72.15 | 61.84 |
| Learning Rate | $2 \times 10^{-5}$ | $5 \times 10^{-6}$ | $1 \times 10^{-6}$ | $5 \times 10^{-7}$ | $2 \times 10^{-5}$ | $5 \times 10^{-6}$ | $1 \times 10^{-6}$ | $5 \times 10^{-7}$ |
| ACC | 94.78 | 94.89 | 93.03 | 91.60 | 94.23 | 94.45 | 93.30 | 91.76 |
| ASR | 100.00 | 99.89 | 94.08 | 60.75 | 86.18 | 77.85 | 61.18 | 42.21 |

Table 4: Comparisons of the defense performance between our proposed method and other defense methods against the word-level, syntactic, add-sentence, and style transfer attacks on SST-2.

| Defense Method | Word-level Attack | | Syntactic Attack | | Add-sentence Attack | | Style Transfer Attack | |
|---|---|---|---|---|---|---|---|---|
| | ACC | ASR | ACC | ASR | ACC | ASR | ACC | ASR |
| ONION | 92.42 | 10.20 | 92.75 | 86.29 | 93.68 | 99.89 | 93.47 | 81.58 |
| BKI | 94.29 | 76.75 | 93.74 | 93.09 | 94.56 | 100.00 | 94.18 | 80.48 |
| STRIP | 94.07 | 99.12 | 93.85 | 89.47 | 94.34 | 100.00 | 94.07 | 85.09 |
| RAP | 94.29 | 82.89 | 93.52 | 91.67 | 93.74 | 87.61 | 86.00 | 85.53 |
| Our Method | 94.23 | **7.89** | 91.98 | **42.11** | 92.81 | **42.21** | 91.76 | **42.21** |

## 4.4 Comparisons with Other Defense Methods

We compare the defense performance of our method with several other defense methods, including ONION (Qi et al., 2021a), Backdoor Keyword Identification (BKI) (Chen and Dai, 2021), STRIP (Gao et al., 2021) and RAP (Yang et al., 2021). For a brief introduction, Backdoor Keyword Identification (BKI) is a training-time defense method by identifying and removing potential poisoned samples from training samples. ONION, STRIP and RAP are inference-time defense methods. We adapt them to the training-time defense for fair comparisons, following (Cui et al., 2022). From the experimental results in Table 4, we can see that the defense performance of our proposed methods is better than other defense methods. The ASR after applying our proposed defense method is lower than those after applying other defense methods.

## 4.5 Experiments on Backdoor Attacks for the Pre-trained CV Model

In this section, we investigate whether our methods could also defend against backdoor attacks for the pre-trained CV model. Specifically, we choose two kinds of backdoor attacks, i.e., inserting patch triggers (Gu et al., 2017) and blending the benign sample with the specific pattern (Chen et al., 2017). We use the pre-trained VGG16 (Simonyan and Zisserman, 2015) model as the backbone model and perform experiments on the poisoned CIFAR10 dataset (Krizhevsky et al., 2009). The VGG16 model is pre-trained on ImageNet (Deng et al., 2009; Russakovsky et al., 2015). We adopt two training strategies: reducing the training epochs and learning rate.

We visualize the changes of ACC and ASR under different settings for two attacks in Figure 6 (a-f). We evaluate both ACC and ASR for each epoch. Our findings are as follows: (1) For the patch-trigger attack, the model mostly learns the original classification task for the first few epochs under a relatively low learning rate, as reflected in the high ACC and low ASR for the first few epochs in Figure 6 (b). This finding seems contradictory to the discovery in Li et al. (2021), which claims that the training loss on poisoned samples declines much faster than that on benign samples in the first few epochs. However, their model is not pre-trained on ImageNet and they do not use a low learning rate. As the model pre-trained on ImageNet has already learned some knowledge about extracting normal features, it is possibly easier for such a model to learn the original classification task. We also explore the effects of pre-training when a language model is trained on a poisoned dataset in appendix B.2. (2) From the comparison between Figure 6 (a) and (c), we can see that jointly reducing the training epochs and learning rate can defend against the patch-trigger attack to some extent, i.e., the ASR drops sharply while the ACC also declines to some degree. (3) Reducing training epochs or learning rate seems not able to defend against the blending-trigger attack as shown in Figure 6 (d, e,

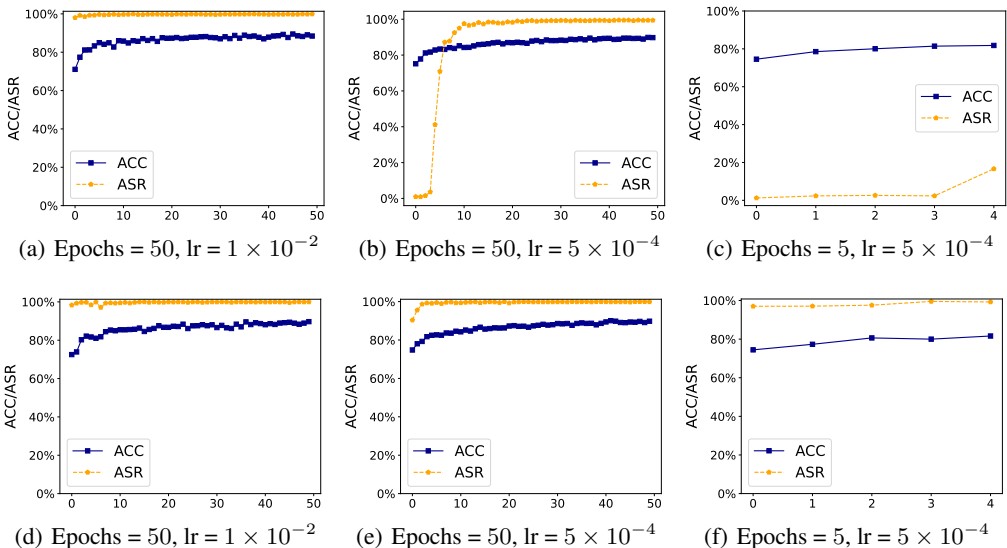

Figure 6: Results of our methods against patch-trigger attack (a, b, c) and blending-trigger attack (d, e, f). We record the changes of ACC and ASR during training when using different training epochs and learning rates.

f). The blending-trigger attack modifies the input data globally rather than alters the input data locally like the patch-trigger attack, which makes the blending-trigger attack harder to defend against.

# 5 Conclusion

In this paper, we reveal the mechanism of the PLM's adaptation on a partially poisoned training dataset. We observe that the model would prioritize learning the common patterns shared by a majority of clean training samples, and then learn the infrequent backdoor patterns corresponding to a small number of poisoned samples. Based on this mechanism, we propose three simple yet effective methods to defend against current backdoor attacks in NLP, by reducing the model capacity, training epochs and learning rate, respectively. We also conduct a series of analyses towards gaining a deeper understanding of the effectiveness of our methods. By rethinking current backdoor attacks in NLP, our findings shed light on how to enhance the security of the model during the PLM's downstream adaptation stage.

# Acknowledgments

This work is supported by the National Key R&D Program of China (No. 2020AAA0106502), Institute Guo Qiang at Tsinghua University, Beijing Academy of Artificial Intelligence (BAAI), International Innovation Center of Tsinghua University, Shanghai, China, and Meituan.

Biru Zhu and Yujia Qin designed the methods and the experiments. Biru Zhu conducted most experiments. Chong Fu conducted the experiments for the pre-trained CV model in § 4.5. Biru Zhu and Yujia Qin wrote the paper. Ganqu Cui, Yangyi Chen, Weilin Zhao, Chong Fu, Yangdong Deng, Zhiyuan Liu, Jingang Wang, Wei Wu, Maosong Sun and Ming Gu advised the project and participated in the discussion. We thank Lifan Yuan for adding the experiments for comparisons with other defense methods in § 4.4 during the rebuttal period.

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
