# OpenReview forum: "Moderate-fitting as a Natural Backdoor Defender for Pre-trained Language Models"
_NeurIPS.cc/2022/Conference — NeurIPS 2022 Accept_

### Official Review · Reviewer_RBg2 · 2022-06-29

**Rating:** 5
**Confidence:** 3
**Soundness:** 2 fair
**Presentation:** 3 good
**Contribution:** 2 fair

**Summary:**

This paper proposes to restrict the Pre-trained Language Models (PLMs)'s adaptation to the moderate-fitting stage to neglect the backdoor triggers for the backdoor defender. Specifically, three methods i.e., reducing the model capacity, training epochs and learning rate are proposed to defend the backdoor attacks. Extensive experiments are conducted on three datasets for two representative backdoor attacks to illustrate the effectiveness of the proposed approach to reduce the impact of the common backdoor attacks against PLM without sacrificing the performance of the model on original data. Furthermore, they also have some experiments to confirm the effectiveness of the proposed approach to other NLP backdoor attacks and to the pre-trained CV models.

**Questions:**

- Line 205 arranges the comparison with the vanilla LoRA and Adapter in Appendix B.4 and B.5, why not put these comparisons in the main body, since they are important to prove the proposed global low-rank approach? Furthermore, can you provide more explanations about why both compared approaches completely fail to defence against backdoor attacks? If they are working for reducing the model capacity, why they are not working for backdoor defenders?
- Why not compare the proposed approach with other backdoor defender techniques? In Section Backdoor defence in NLP, there are some different techniques, why not take them as the baselines? There are also some related works that are missed in this paper for example PICCOLO [1].

[1]. PICCOLO : Exposing Complex Backdoors in NLP Transformer Models. Liu et al.



**Limitations:**

- The technique novelty is not enough and lacks the comparison with some state-of-the-art backdoor defender techniques.

**Strengths And Weaknesses:**

Strengths:
- The paper is well-written and clear in its purpose and objectives, and it is easy to follow for a non-specialist.
- The PET algorithm of global low-rank proposed in this paper is interesting, and reasonable and effective results are obtained in the experiment. There have been plenty of experiments and analyses to validate the proposed approach

Weakness:
- The technical novelty is not enough, the proposed second and third approaches (reducing training epochs and learning rate) are too simple to take as a methodology. Although they have proved the effectiveness in preventing the backdoor attack in the paper, they are more seems to the hyper-parameter tuning and both are tricks for the model training.
- This paper validates the proposed approach comprehensively, however, it lacks the comparison with some state-of-the-art backdoor defender techniques to prove that they can achieve the new state-of-the-art performance.

---

> ### Author Response · Authors · 2022-08-01
> **Response 3 to Reviewer RBg2**
>
> **Comment 3:** ***"Line 205 arranges the comparison with the vanilla LoRA and Adapter in Appendix B.4 and B.5, why not put these comparisons in the main body, since they are important to prove the proposed global low-rank approach? Furthermore, can you provide more explanations about why both compared approaches completely fail to defence against backdoor attacks? If they are working for reducing the model capacity, why they are not working for backdoor defenders?''***
>
> **Response:**  Thank you for the valuable suggestion. We will put these comparisons in the main body to prove the proposed global low-rank approach. Though vanilla LoRA and Adapter reduce the model capacity, however, the extent of such reduction is still far from causing moderate-fitting, as stated in line 133-134 in the manuscript.
>
> ----------
>
> **Comment 4:** ***"Why not compare the proposed approach with other backdoor defender techniques? In Section Backdoor defence in NLP, there are some different techniques, why not take them as the baselines? There are also some related works that are missed in this paper for example PICCOLO [1].''***
>
> **Response:** Thank you for the constructive suggestion. (1) Following your suggestion, we have compared our proposed method with other backdoor defense techniques [2,3,4,5]. The experimental results are shown in Table 3 and Table 4. From the experimental results, we can see that the defense performance of our proposed method is **better** than other defense methods. The **ASR** after applying our proposed defense method is **lower** than those after applying other defense methods.
>
> | Defender               | ONION  | BKI   | STRIP  | RAP    | Our Method |
> | ---------------------- | ------ | ----- | ------ | ------ | ---------- |
> | Word-level (ACC)     | 92.42  | 94.29 | 94.07  | 94.29  | 94.23      |
> | Word-level (ASR)     | 10.20  | 76.75|  99.12 |  82.89 | **7.89**       |
> | Syntactic (ACC)      |  92.75 | 93.74 |93.85   |  93.52 | 91.98      |
> | Syntactic (ASR)      | 86.29 | 93.09 |  89.47 | 91.67  | **42.11**      |
> | Add-sentence (ACC)   | 93.68  | 94.56 | 94.34  | 93.74  | 92.81      |
> | Add-sentence (ASR)   |99.89  | 100.00 | 100.00 |87.61  | **42.21**      |
> | Style-Transfer (ACC) | 93.47 | 94.18 | 94.07  |  86.00 | 91.76      |
> | Style-Transfer (ASR) | 81.58  |80.48  | 85.09 |  85.53 | **42.21**      |
>
> Table 3: Comparisons of the defense performance between our proposed method and other defense methods against the word-level, syntactic, add-sentence, and style transfer attacks on SST-2.
>
> |  Defender  | ACC | ASR |
> | :--------: | :--------------: | :--------------: |
> |   ONION    |   90.28         |      99.35       |
> |    BKI     |      90.76      |     99.47       |
> |   STRIP    |        91.11     |      99.44       |
> |    RAP     |     90.45    |      99.67       |
> | Our Method |      88.45       |      **67.14**       |
>
> Table 4: Comparisons of the defense performance between our proposed method and other defense methods against the syntactic attack on AG News.
>
> (2) PICCOLO targets at distinguishing trojaned models from clean models. PICCOLO's threat model is different from ours. In PICCOLO, they assume the attacker has full control of the training process. The defender is given a model and a few clean sentences to determine if the model contains backdoor. In our setting, the attacker only poisons the training data and can not control the model training process. The defender aims to train a backdoor-free model with the downloaded third-party poisoned dataset. Although our threat model is different from PICCOLO, PICCOLO is indeed an important related work. We will cite PICCOLO in the related work (Backdoor Defense in NLP) and illustrate the difference between our threat model and theirs in the revised paper.
>
> ----------
>
> **References:**
>
> [1] Piccolo: Exposing complex backdoors in nlp transformer models. In 2022 IEEE Symposium on Security and Privacy (SP).
>
> [2]  Onion: A simple and effective defense against textual backdoor attacks. In EMNLP 2021.
>
> [3] Mitigating backdoor attacks in lstm-based text classification systems by backdoor keyword identification. Neurocomputing, 2021.
>
> [4] Design and evaluation of a multi-domain trojan detection method on deep neural networks. IEEE Transactions on Dependable and Secure Computing, 2021.
>
> [5] Rap: Robustness-aware perturbations for defending against backdoor attacks on nlp models. In EMNLP 2021.

---

> ### Author Response · Authors · 2022-08-01
> **Response 2 to Reviewer RBg2**
>
> **Comment 2:** ***"This paper validates the proposed approach comprehensively, however, it lacks the comparison with some state-of-the-art backdoor defender techniques to prove that they can achieve the new state-of-the-art performance.''***
>
> **Response:** Thank you for the valuable suggestion.  (1) Firstly, following your suggestion, we have compared the defense performance of our method with other backdoor defense methods, including ONION [1], Backdoor Keyword Identification (BKI) [2], STRIP [3] and RAP [4]. For a brief introduction, Backdoor Keyword Identification (BKI) is a training-time defense method by identifying and filtering out poisoning samples from training samples. ONION, STRIP and RAP are inference-time defense methods. We adapt them to the training-time defense for comparison. The experimental results are shown in Table 1 and Table 2. From the experimental results, we can see that the defense performance of our proposed methods is **better** than other defense methods. The **ASR** after applying our proposed defense method is **lower** than those after applying other defense methods. For the syntactic-level attack, the defense performance of other methods is much poorer than ours.
>
> | Defender               | ONION  | BKI   | STRIP  | RAP    | Our Method |
> | ---------------------- | ------ | ----- | ------ | ------ | ---------- |
> | Word-level (ACC)     | 92.42  | 94.29 | 94.07  | 94.29  | 94.23      |
> | Word-level (ASR)     | 10.20  | 76.75|  99.12 |  82.89 | **7.89**       |
> | Syntactic (ACC)      |  92.75 | 93.74 |93.85   |  93.52 | 91.98      |
> | Syntactic (ASR)      | 86.29 | 93.09 |  89.47 | 91.67  | **42.11**      |
> | Add-sentence (ACC)   | 93.68  | 94.56 | 94.34  | 93.74  | 92.81      |
> | Add-sentence (ASR)   |99.89  | 100.00 | 100.00 |87.61  | **42.21**      |
> | Style-Transfer (ACC) | 93.47 | 94.18 | 94.07  |  86.00 | 91.76      |
> | Style-Transfer (ASR) | 81.58  |80.48  | 85.09 |  85.53 | **42.21**      |
>
> Table 1: Comparisons of the defense performance between our proposed method and other defense methods against the word-level, syntactic, add-sentence, and style transfer attacks on SST-2.
>
> |  Defender  | ACC | ASR |
> | :--------: | :--------------: | :--------------: |
> |   ONION    |   90.28         |      99.35       |
> |    BKI     |      90.76      |     99.47       |
> |   STRIP    |        91.11     |      99.44       |
> |    RAP     |     90.45    |      99.67       |
> | Our Method |      88.45       |      **67.14**       |
>
> Table 2: Comparisons of the defense performance between our proposed method and other defense methods against the syntactic attack on AG News.
>
> (2) Secondly, our defense methods are **orthogonal** to the other defense methods, and can be used together with other methods. Other defense methods either filter training samples or testing samples. For the training-time defense, the victim can first filter the training samples and then use our backdoor-free training method to train the model on the filtered training dataset. For the inference-time defense, after training the model using our backdoor-free training method, the victim can further perform inference on the filtered testing dataset. We leave the combination of our proposed backdoor-free training method and other defense methods in future work.
>
> ----------
>
> **References:**
>
> [1]  Onion: A simple and effective defense against textual backdoor attacks. In EMNLP 2021.
>
> [2] Mitigating backdoor attacks in lstm-based text classification systems by backdoor keyword identification. Neurocomputing, 2021.
>
> [3] Design and evaluation of a multi-domain trojan detection method on deep neural networks. IEEE Transactions on Dependable and Secure Computing, 2021.
>
> [4] Rap: Robustness-aware perturbations for defending against backdoor attacks on nlp models. In EMNLP 2021.

---

> ### Author Response · Authors · 2022-08-01
> **Response 1 to Reviewer RBg2**
>
> We thank the reviewer for the insightful and constructive feedback for improving this paper. Please find below our point-to-point response to your comments.
>
> **Comment 1:** ***"The technical novelty is not enough, the proposed second and third approaches (reducing training epochs and learning rate) are too simple to take as a methodology. Although they have proved the effectiveness in preventing the backdoor attack in the paper, they are more seems to the hyper-parameter tuning and both are tricks for the model training.''***
>
> **Response:**
>
> **We believe simplicity is actually an advantage of our method.** Following the Occam’s razor principle, we do not pursue designing a sophisticated algorithm, instead, we propose a **simple and effective** one.
>
> The novelty of this paper is as follows.
>
> (1) To the best of our knowledge, this is the **first work** to explore the possibility of backdoor-free adaptation for PLMs.
>
> (2) We revealed the mechanism of distinct learning phases for the backdoor task and original task during PLM's adaptation on a poisoned dataset.
>
> (3) We design three methods to defend against backdoor attacks by reducing the model capacity, training epochs, and learning rate, respectively. For reducing the model capacity, we propose a novel global low-rank architecture which is applied to PET (parameter-efficient tuning) algorithms.
>
> (4) We also analyze the reason why our method works and show the visualization of PLM's learning dynamics in Section 4.2.

---

> ### Author Response · Authors · 2022-08-05
> **Looking forward to your reply. (to Reviewer RBg2)**
>
> Dear reviewer,
>
> Thank you for your valuable suggestions and constructive comments. We have done the point-to-point response to your comments. Could you please let us know whether our responses address your concerns? We would really appreciate it if you could let us know before the reviewer-author discussion period ends.

---

> > ### Comment · Reviewer_RBg2 · 2022-08-08
> > **Still wondering the effectiveness of  global-rank.**
> >
> > Thanks for the very detailed responses. I agree that simplicity is an advantage of your method and thanks for adding the relevant baselines for comparison. But I still wondering why vanilla LoRA and Adapter fail to defence against backdoor attacks. Currently, the explanaion of Adapter failed is due to the low-rank constraint, but when adding the low-rank (i.e.,, LoRA), it still fails and maybe it due to the global-rank. But can you provide more insights for that to convince me the effective of global-rank.

---

> > > ### Author Response · Authors · 2022-08-08
> > > **Response to Reviewer RBg2**
> > >
> > > Thank you for your inspiring comments and valuable suggestions.
> > >
> > > The low-rank structures of the vanilla LoRA are individually distributed in different Transformer layers. Even if each local structure has a low-rank restriction, the overall rank of all the local structures can still be high. This is the reason why the vanilla LoRA cannot successfully constrain the overall model capacity; instead, we argue that the model capacity of a PLM is determined by the global intrinsic rank of the tunable parameters. Therefore, it is essential to constrain the overall intrinsic rank of the weight updates to be lower than a threshold (as our proposed global low-rank method does).
> > >
> > > We have performed experiments to demonstrate this viewpoint. The experimental results of reducing the local rank *r* of LoRA against the word-level attack are shown in Table 1 (in Figure 1 in the manuscript). The experimental results of reducing the global rank of our proposed reparameterized LoRA against the word-level attack are shown in Table 2 (in Table 5 in the appendix).
> > >
> > > From the experimental results in Table 1, we can see that even if the local rank *r* of LoRA is extremely low (reduced to 1), the ASR is still very high (96.82\%). Also, we can find that reducing local rank $r$ of vanilla LoRA does not have much influence on the ACC and ASR, which demonstrates the defects of the local low-rank architecture of vanilla LoRA when defending against backdoor attacks.
> > >
> > > However, if we reduce the global rank of our proposed reparameterized LoRA, the ASR drops sharply.  As shown in Table 2, when the bottleneck dimension changes from 256 to 1, the ASR decreases from 98.79\% to 10.96\%. The experimental results demonstrate the effectiveness of our proposed global low-rank reparameterization network.  We will explain it more explicitly in the revision.
> > >
> > > | local rank *r* |  16   |   8   |   4   |   1   |
> > > | :------------: | :---: | :---: | :---: | :---: |
> > > |  ACC (SST-2)   | 94.45 | 94.29 | 94.34 | 94.56 |
> > > |  ASR (SST-2)   | 96.05 | 96.16 | 95.50 | **96.82** |
> > >
> > > Table 1:Results of the vanilla  LoRA when using different local rank *r*
> > >
> > > | global rank (*bottleneck dimension*) |  256  |  32   |   4   |   2   |   1   |
> > > | :------------: | :---: | :---: | :---: | :---: | :---: |
> > > |  ACC (SST-2)   | 94.78 | 95.00 | 93.96 | 92.59 | 92.64 |
> > > |  ASR (SST-2)   | 98.79 | 98.36 | 11.95 | 11.07 | **10.96** |
> > >
> > > Table 2: Results of our proposed reparameterized  LoRA when using different global rank (*bottleneck dimension*)

---

> > > > ### Comment · Reviewer_RBg2 · 2022-08-08
> > > > **Update the score from 4 to 5.**
> > > >
> > > > Thanks for answering our question. After reading the author's response and other reviewers' comments. I am willing to improve my score from 4 to 5.

---

> > > > > ### Author Response · Authors · 2022-08-08
> > > > > **Response to Reviewer RBg2**
> > > > >
> > > > > We sincerely thank you for your effort to improve this paper and your decision.

---

### Official Review · Reviewer_eiAZ · 2022-07-05

**Rating:** 6
**Confidence:** 5
**Soundness:** 3 good
**Presentation:** 3 good
**Contribution:** 2 fair

**Summary:**

This paper proposes a method to defend against training data poisoning attacks on pre-trained language models (PLMs). The proposed method, moderate-fitting, is based on the observation that during fine-tuning, the PLM will first learn the major features in the training dataset, and then starts to fit on the poisoned samples. They propose to (1) reduce the model capacity using low-rank parameter-efficient tuning (PET) (2) reduce the training epochs (3) lower the learning rate to prevent the PLM from fitting on the poisoned dataset.
The proposed method is shown to be highly effective against word-level backdoor triggers on three different text classification datasets.

**Questions:**

### Questions
1. In Figure 5, is the ASR/ACC and tSNE the result of the training set or the testing set? I have this question because if we want to know how well the model "fits", we normally mean "how well the model fits on the training data".
2. In the experiment of the synthetic dataset, why not use the whole dataset of SST2? In the experiment in Appendix B.7, why not use the whole AG-News dataset?
3. What does Line 10 in the Appendix mean? Does it mean that only the 11106 samples are used for fine-tuning, and 10% of the 11106 samples are poisoned? If this is the case, why is this so?
4. I am not convinced by using training epochs as early stopping. One epoch for a large dataset and one epoch for a small dataset is very different, and it is also possible that for a rather easy but large dataset, the over-fitting phase will occur in less than one epoch.

### Suggestions
1. I would recommend first showing that the two phases (moderate-fitting and over-fitting) exist during fine-tuning by plotting something like Figure 5 (but with standard fine-tuning) before introducing the method in Section 3.
2. I suggest testing the proposed method on different sizes and types of PLMs, instead of only using RoBERTa-BASE.
3. I suggest using more diverse datasets. The three datasets used in this paper are quite simple classification tasks that can be categorized by superficial word-level clues. I would like to know whether the proposed method will still be effective for tasks such as natural language understanding, e.g., QNLI, MNLI, and RTE.

**Limitations:**

The authors have addressed the limitations and the ethical concerns.

**Strengths And Weaknesses:**

Update during author/reviewer discussion period
===
Since all my questions and concerns are properly addressed, I raise the score to 6.
***
***


### Strength
1. The paper is overall well-written and easy to follow
2. The proposed method can effectively defend against word-level attacks on three different datasets.

### Weakness
Please correct me if I am misunderstanding any part of the paper.
1. No baseline methods are included for comparison, and it is hard to understand how effective is the proposed method compared to other defense methods. *(This is addressed during rebuttal)*
    1. For the word-level backdoor attack, the performance of the proposed method should be compared with other baseline methods, such as ONION.
    2. When defending the syntactic-level backdoor attack, the attack success rate (ASR) of the proposed method is not very low. Without comparing it with other defense methods, it is hard to tell how effective the proposed method is.
2. The performance of all the three proposed method on AG-News when defending against syntactic attacks are not very convincing. This makes me doubt whether the proposed methods are general enough for other datasets.
3. Some experiment settings are not very clear. *(These are addressed during rebuttal)* For example:
    1. For SST-2, how is the data split? Is the ASR shown for SST-2 the performance on the official testing set or the development set? In Appendix A.1, it says the accuracy of SST-2 is calculated based on the official testing dataset. But since the label of the testing dataset of SST-2 is not publicly available, it is unclear how the ASR on SST-2 is calculated.
    2. How many samples are used when training for each of the three datasets? This matters since 1 epoch for a large dataset is different from 1 epoch for a small dataset.

---

> ### Author Response · Authors · 2022-08-01
> **Response 5 to Reviewer eiAZ**
>
> **Comment 6:** ***What does Line 10 in the Appendix mean? Does it mean that only the 11106 samples are used for fine-tuning, and 10\% of the 11106 samples are poisoned? If this is the case, why is this so?''***
>
> **Response:** Yes, it means that only the 11106 samples are used for fine-tuning, and 10\% of the 11106 samples are poisoned. The sampled AG News dataset we used follows the sampled AG News dataset used in the previous work [1].
>
> ----------
>
> **Comment 7:** ***"I am not convinced by using training epochs as early stopping. One epoch for a large dataset and one epoch for a small dataset is very different, and it is also possible that for a rather easy but large dataset, the over-fitting phase will occur in less than one epoch.''***
>
> **Response:** Thank you for the constructive suggestion. For large datasets, we can use training steps as the criterion for early stopping, as PLMs generally require some training steps to fit a large dataset. Specifically, we can see the clean accuracy on the validation dataset with a fixed interval of training steps to determine when to do the early stop. If the accuracy on the validation dataset is high enough at a certain training step, we can early stop the training process.
>
> ----------
>
> **Comment 8:** ***"I would recommend first showing that the two phases (moderate-fitting and over-fitting) exist during fine-tuning by plotting something like Figure 5 (but with standard fine-tuning) before introducing the method in Section 3.''***
>
> **Response:**  Thank you for the constructive suggestion. Following your suggestion, we have plotted the PLM's learning dynamics during standard fine-tuning and put the figure at https://www.dropbox.com/s/o6t1ob8f26wteqj/Visualization%20of%20PLM%E2%80%99s%20learning%20dynamics%20when%20using%20the%20standard%20fine-tuning%20method.png?dl=0.
>
> It shows that the two phases (moderate-fitting and over-fitting) exist during standard fine-tuning. We will put the figure before introducing the method in Section 3 in the revised paper.
>
> ----------
>
> **Comment 9:** ***"I suggest testing the proposed method on different sizes and types of PLMs, instead of only using RoBERTa-BASE.''***
>
> **Response:** Thank you for the constructive suggestion. Following your suggestion, we have performed experiments on RoBERTa-LARGE and BERT-BASE models. We fine-tune the RoBERTa-LARGE and BERT-BASE models with the learning rate of $2\times10^{-5}$ under different training epochs on SST-2. The results are shown in Table 6 and Table 7, respectively. From the experimental results, we can see that our proposed method is effective for different sizes and types of PLMs. These results demonstrate that our moderate-fitting can be applied to various PLMs. We will add these results to the revised paper.
>
> |   Epochs   |  10   |   2   |   1   |
> | :--------: | :---: | :---: | :---: |
> | SST-2 (ACC) | 95.55 | 95.83 | 95.50 |
> | SST-2 (ASR) | 99.67 | 72.48 | 7.35  |
>
> Table 6: Results of reducing the training epochs against word-level attack when fine-tuning the RoBERTa-LARGE model.
>
>
>
> |   Epochs   |  10   |   2   |   1   |
> | :--------: | :---: | :---: | :---: |
> | SST-2 (ACC) | 91.93 | 91.32 | 90.66 |
> | SST-2 (ASR) | 99.67 | 90.57 | 18.86 |
>
> Table 7: Results of reducing the training epochs against word-level attack when fine-tuning the BERT-BASE model.
>
> ----------
>
> **Comment 10:** ***"I suggest using more diverse datasets. The three datasets used in this paper are quite simple classification tasks that can be categorized by superficial word-level clues. I would like to know whether the proposed method will still be effective for tasks such as natural language understanding, e.g., QNLI, MNLI, and RTE.''***
>
> **Response:** We thank the reviewer for the valuable suggestion. Following your suggestion, we have performed new experiments on QNLI against the word-level attack. From the original QNLI training dataset, we sample 80\% as our training dataset and sample 10\% as the testing dataset. The results are shown in Table 8. Our proposed method can be applied to different NLP tasks, including the natural language understanding task.
>
> | Bottleneck Dimension |   4   |   2   |   1   |
> | :------------------: | :---: | :---: | :---: |
> |      QNLI (ACC)      | 86.66 | 84.89 | 82.68 |
> |      QNLI (ASR)      | 97.01 | 28.33 | 19.94 |
>
> Table 8: Results of reducing the model capacity using reparameterized LoRA against word-level attack on QNLI.
>
> ----------
>
> **References:**
>
> [1] Mind the style of text! adversarial and backdoor attacks based on text style transfer. In EMNLP 2021.

---

> > ### Comment · Reviewer_eiAZ · 2022-08-04
> > **Updating score to 6**
> >
> > Thanks to the author for the very detailed responses, and they successfully answer all my questions and my concerns. I raise my score to 6. I hope the authors will add those baseline performances to the revision soon. As an important future work, I think it is important to discuss how the truncated training steps undermine the model's performance on dimensions other than backdoor attack. For example, will the model trained with lesser steps be less robust to OOD samples? Or will the model perform worse on those samples falling in the long-tail part of the data distribution? Or will the model be less prone to synonym substitution attacks? Still, I think those topics I mentioned above are out of the scope of the paper, and need not be addressed in this paper.
> >
> > A very minor point is in Table 8 of response 5, the SST-2 should be QNLI.

---

> > > ### Author Response · Authors · 2022-08-04
> > > **Response to Reviewer eiAZ**
> > >
> > > Thank you for your valuable suggestions and feedback. We will add those baseline performances in the revision. We also admit that it is interesting to discuss how the truncated training steps influence the model's performance on other dimensions. We will further explore this problem in future work. By the way, following your suggestion, we have updated and corrected the table header of Table 8.

---

> ### Author Response · Authors · 2022-08-01
> **Response 4 to Reviewer eiAZ**
>
> **Comment 3:** ***"Some experiment settings are not very clear. For example: (1) For SST-2, how is the data split? Is the ASR shown for SST-2 the performance on the official testing set or the development set? In Appendix A.1, it says the accuracy of SST-2 is calculated based on the official testing dataset. But since the label of the testing dataset of SST-2 is not publicly available, it is unclear how the ASR on SST-2 is calculated. (2) How many samples are used when training for each of the three datasets? This matters since 1 epoch for a large dataset is different from 1 epoch for a small dataset.''***
>
> **Response:** (1) Following [1], we use the data split of the original version of SST-2 dataset in [2], instead of the GLUE-version [3] SST-2 dataset. The original version of SST-2 dataset contains the labeled testing samples; while the GLUE version of SST-2 does not contain publicly available labels for testing samples.
>
> (2) For SST-2, HSOL, and AG News, 6920, 5832, and 11106 samples are used for training the model, respectively. We will add the description of the number of training samples for each of the three datasets in the revised paper.
>
> ----------
>
> **Comment 4:** ***"In Figure 5, is the ASR/ACC and tSNE the result of the training set or the testing set? I have this question because if we want to know how well the model "fits", we normally mean "how well the model fits on the training data".''***
>
> **Response:** In Figure 5, the ASR/ACC is the result of the testing set. The tSNE is the result of the development set.
> The development set and test set follow the same data distribution as the training set.
>
> Following your suggestion, we have performed new experiments to see the ASR/ACC of the training set. The ACC is tested on the clean training dataset part. The ASR is tested on the poisoned training dataset part. The results are shown in the new figure at https://www.dropbox.com/s/5j2j8vsbfz32fia/Visualization%20of%20the%20changes%20of%20ACC%20and%20ASR%20on%20the%20training%20dataset..png?dl=0.
>
> From the new figure and the original Figure 5 in the manuscript, we can see that the trend of changes of ACC/ASR on the training dataset and testing dataset are similar. We will add these results to the revised paper.
>
> ----------
>
> **Comment 5:** ***"In the experiment of the synthetic dataset, why not use the whole dataset of SST2? In the experiment in Appendix B.7, why not use the whole AG-News dataset?''***
>
> **Response:** In the experiment of the synthetic dataset in the main paper, we have used the whole dataset of SST-2. We use the original version of SST-2 [2], which follows the dataset used in the previous work [1]. In the experiment in Appendix B.7, we take all samples whose labels are “World” or “Sports” from our originally used AG News training dataset. Our originally used AG News training dataset follows the dataset used in the previous work [1].
>
> ----------
>
> **References:**
>
> [1] Mind the style of text! adversarial and backdoor attacks based on text style transfer. In EMNLP 2021.
>
> [2] Recursive deep models for semantic compositionality over a sentiment treebank. In EMNLP 2013.
>
> [3] Glue: A multi-task benchmark and analysis platform for natural language understanding. arXiv preprint arXiv:1804.07461, 2018.

---

> ### Author Response · Authors · 2022-08-01
> **Response 3 to Reviewer eiAZ**
>
> **Comment 2:** ***"The performance of all the three proposed method on AG-News when defending against syntactic attacks are not very convincing. This makes me doubt whether the proposed methods are general enough for other datasets.''***
>
> **Response:** (1) Our proposed method is general for various datasets. We have performed experiments on three representative datasets, i.e., SST-2 [1], AG News [2] and Hate Speech and Offensive Language (HSOL) [3], which are commonly used datasets in the field of backdoor attack/defense in NLP [4,5,6]. To address your concerns, we further perform new experiments on the rotten tomatoes [7] dataset. Specifically, we defend against syntactic attack using reparameterized LoRA tuning with the small bottleneck dimension. The experimental results are shown in Table 4. From the experimental results, we can see that the ASR declines from **87.80\%** to **42.96\%** when the bottleneck dimension changes from 32 to 1. However, the ACC does change much. These results demonstrate that our proposed method can be applied to various datasets.
>
> | Bottleneck Dimension |  32   |   4   |   2   |   1   |
> | :------------------: | :---: | :---: | :---: | :---: |
> |       Rotten Tomatoes  (ACC)     | 86.96 | 86.59 | 87.24 | 86.87 |
> |       Rotten Tomatoes  (ASR)      | 87.80 | 69.79 | 51.97 | 42.96 |
>
> Table 4: Results of reducing the model capacity using reparameterized LoRA against syntactic attack on the rotten tomatoes dataset.
>
> (2) Syntactic attack on AG News is also difficult to defend against for other defense methods as shown in Table 5. The performance of our proposed defense method is **better** than other defense methods when defending against syntactic attacks on AG News.
>
> |  Defender  | ACC | ASR |
> | :--------: | :--------------: | :--------------: |
> |   ONION    |   90.28         |      99.35       |
> |    BKI     |      90.76      |     99.47       |
> |   STRIP    |        91.11     |      99.44       |
> |    RAP     |     90.45    |      99.67       |
> | Our Method |      88.45       |      **67.14**       |
>
> Table 5: Comparisons of the defense performance between our proposed method and other defense methods against the syntactic attack on AG News.
>
> ----------
>
> **References:**
>
> [1]  Recursive deep models for semantic compositionality over a sentiment treebank. In EMNLP 2013.
>
> [2] Character-level convolutional networks for text classification. In NIPS 2015.
>
> [3] Automated hate speech detection and the problem of offensive language.  In Proceedings of the 11th International AAAI Conference on Web and Social Media, ICWSM ’17, 2017.
>
> [4] Mind the style of text! adversarial and backdoor attacks based on text style transfer. In EMNLP 2021.
>
> [5] Hidden killer: Invisible textual backdoor attacks with syntactic trigger. In ACL 2021.
>
> [6] Onion: A simple and effective defense against textual backdoor attacks. In EMNLP 2021.
>
> [7] Seeing stars: Exploiting class relationships for sentiment categorization with respect to rating scales. In ACL 2005.

---

> ### Author Response · Authors · 2022-08-01
> **Response 2 to Reviewer eiAZ**
>
> **Comment 1.2:** ***"When defending the syntactic-level backdoor attack, the attack success rate (ASR) of the proposed method is not very low.''***
>
> **Response:** Firstly, the high ASR of the backdoor attacks using syntactic triggers may be attributed to the reason that such triggers **change the semantic information of the text sample dramatically** [1]. It is even possible that the syntactic paraphrase **changes the ground-truth label of texts**. For example, one original sentence from SST-2 is "neither funny nor suspenseful nor particularly well-drawn", which is labeled as negative. After using the syntactic paraphrase in [2], it is transformed into "when it 's funny , it 's nice and tight", which is near positive. Therefore, the model is reasonable to "misclassify" samples with such triggers, which is irrelevant to backdoor attacks.
>
> To further prove this viewpoint, we have performed experiments to see the ASR of the syntactic and word-level triggers on the clean models trained under two settings. We fine-tune the RoBERTa-BASE model on SST-2 with 10 epochs. We also use reparameterized LoRA with the bottleneck dimension 1 to train the model. The experimental results are shown in Table 3. From the experimental results, we can see that even on clean models, the ASR of the syntactic trigger may be above 20\%, which is significantly higher than the word-level trigger. To sum up, the success of syntactic triggers is with the cost of significantly changing or even flipping the semantic information of the samples. Thus, the high ASR value after defense does not indicate that our defense method is ineffective.
>
> |    Trigger Type     | Word-level  ACC | Word-level ASR | Syntactic ACC | Syntactic ASR |
> | :-----------------: | :---------------: | :---------------: | :--------------: | :--------------: |
> | Finetune |       94.23       |       **7.13**        |      94.23       |      **19.96**       |
> |        LoRA         |       92.97       |       **9.76**        |      92.97       |      **20.83**       |
>
> Table 3: The ACC and ASR after training the model with clean training data on SST-2 for word-level triggers and syntactic triggers.
>
> ----------
>
> **References:**
>
> [1] Rethink the evaluation for attack strength of backdoor attacks in natural language processing.
>
> [2] Hidden killer: Invisible textual backdoor attacks with syntactic trigger. In ACL 2021

---

> ### Author Response · Authors · 2022-08-01
> **Response 1 to Reviewer eiAZ**
>
> We thank the reviewer for the insightful and constructive feedback for improving this paper. Please find below our point-to-point response to your comments.
>
> **Comment 1.1:** ***"No baseline methods are included for comparison, and it is hard to understand how effective is the proposed method compared to other defense methods."***
>
> **Response:** Thank you for the valuable suggestion.  (1) Firstly, following your suggestion, we have compared the defense performance of our method with other backdoor defense methods, including ONION [1], Backdoor Keyword Identification (BKI) [2], STRIP [3] and RAP [4]. For a brief introduction, Backdoor Keyword Identification (BKI) is a training-time defense method by identifying and filtering out poisoning samples from training samples. ONION, STRIP and RAP are inference-time defense methods. We adapt them to the training-time defense for comparison. The experimental results are shown in Table 1 and Table 2. From the experimental results, we can see that the defense performance of our proposed methods is **better** than other defense methods. The **ASR** after applying our proposed defense method is **lower** than those after applying other defense methods. For the syntactic-level attack, the defense performance of other methods is much poorer than ours.
>
> | Defender               | ONION  | BKI   | STRIP  | RAP    | Our Method |
> | ---------------------- | ------ | ----- | ------ | ------ | ---------- |
> | Word-level (ACC)     | 92.42  | 94.29 | 94.07  | 94.29  | 94.23      |
> | Word-level (ASR)     | 10.20  | 76.75|  99.12 |  82.89 | **7.89**       |
> | Syntactic (ACC)      |  92.75 | 93.74 |93.85   |  93.52 | 91.98      |
> | Syntactic (ASR)      | 86.29 | 93.09 |  89.47 | 91.67  | **42.11**      |
> | Add-sentence (ACC)   | 93.68  | 94.56 | 94.34  | 93.74  | 92.81      |
> | Add-sentence (ASR)   |99.89  | 100.00 | 100.00 |87.61  | **42.21**      |
> | Style-Transfer (ACC) | 93.47 | 94.18 | 94.07  |  86.00 | 91.76      |
> | Style-Transfer (ASR) | 81.58  |80.48  | 85.09 |  85.53 | **42.21**      |
>
> Table 1: Comparisons of the defense performance between our proposed method and other defense methods against the word-level, syntactic, add-sentence, and style transfer attacks on SST-2.
>
> |  Defender  | ACC | ASR |
> | :--------: | :--------------: | :--------------: |
> |   ONION    |   90.28         |      99.35       |
> |    BKI     |      90.76      |     99.47       |
> |   STRIP    |        91.11     |      99.44       |
> |    RAP     |     90.45    |      99.67       |
> | Our Method |      88.45       |      **67.14**       |
>
> Table 2: Comparisons of the defense performance between our proposed method and other defense methods against the syntactic attack on AG News.
>
> (2) Secondly, our defense methods are **orthogonal** to other defense methods, and can be used together with other methods. Other defense methods either filter training samples or testing samples. For the training-time defense, the victim can first filter the training samples and then use our backdoor-free training method to train the model on the filtered training dataset. For the inference-time defense, after training the model using our backdoor-free training method, the victim can further perform inference on the filtered testing dataset. We leave the combination of our proposed backdoor-free training method and other defense methods in future work.
>
> ----------
>
> **References:**
>
> [1]  Onion: A simple and effective defense against textual backdoor attacks. In EMNLP 2021.
>
> [2] Mitigating backdoor attacks in lstm-based text classification systems by backdoor keyword identification. Neurocomputing, 2021.
>
> [3] Design and evaluation of a multi-domain trojan detection method on deep neural networks. IEEE Transactions on Dependable and Secure Computing, 2021.
>
> [4] Rap: Robustness-aware perturbations for defending against backdoor attacks on nlp models. In EMNLP 2021.

---

### Official Review · Reviewer_9wod · 2022-07-11

**Rating:** 6
**Confidence:** 4
**Soundness:** 2 fair
**Presentation:** 2 fair
**Contribution:** 2 fair

**Summary:**

This paper proposed a defense method named moderate-fitting against NLP backdoor attacks for pre-trained language models. The key observation is that during fine-tuning on the poisoned dataset, the PLM follows two learning stages: the moderate-fitting stage, which mainly focuses on learning major features(i.e. clean samples), and an overfitting stage, which learns subsidiary features(i.e. trigger features). Based on the observation, the authors proposed 3 simple training strategies to reduce models’ capacities, hence they will stay in the moderate-fitting stage and not learn the backdoor features. Evaluations on 4 different NLP attacks and 3 text classification datasets demonstrated the effectiveness of the proposed method.

**Questions:**

Please refer Strengths and Weaknesses

**Strengths And Weaknesses:**

- Strengths
    - The topic is very interesting and critical for the community.
    - The idea is simple and easy to understand.
    - The evaluation is overall comprehensive. Authors evaluated their proposed methods under various types of NLP attacks on several NLP datasets. The evaluation results demonstrated the effectiveness of their proposed method on certain types of attacks.

- Weaknesses
    - The proposed method shares similarity with several existing works in vision domains. The overreaching goal of this work is to train a clean model even if the dataset is poisoned. There are several similar works for the vision model backdoor defense[1-2]. Although they were originally proposed for vision models. I would like to see more discussion about the possibility to extend such methods for PLMs and compare them with the proposed method. It will be more convincing if the proposed method outperforms them.
    - The paper organization should be further improved. For example, the authors did not provide the threat model explicitly to describe the attacker and defender’s capability. Based on my understanding, the authors assumed the poisoning happens during the fine-tuning stage, i.e. the attackers poisoned the dataset used in the fine-tuning. If so, such a scenario seems too narrow for me. What if the attack happens during the pre-training stage? Or users want to train their models from scratch. Under such situations, it’s unknown whether the model’s training still strictly follows moderate-fitting and overfitting stages. If not, please further clarify the threat model in the revision.
    - The proposed method seems not very effective against several advanced NLP backdoor attacks. As shown in Fig.2-4, Table.1,3. The syntactic, sentence and style-based triggers remain effective after the defense (over 50% ASR after defense), which might indicate the proposed method is not general and can not handle more stealthy triggers than simple word-level triggers.
    - When evaluating the proposed method on several vision backdoor attacks, why only report the evaluation results of altering learning rate and training epochs?  What’s the performance of the proposed reparameterized PET on such vision models?

------------------
Update: missing references

[1] Huang, Kunzhe, et al. "Backdoor defense via decoupling the training process." arXiv preprint arXiv:2202.03423 (2022)
[2] Li, Yige, et al. "Anti-backdoor learning: Training clean models on poisoned data." Advances in Neural Information Processing Systems 34 (2021): 14900-14912.

---

> ### Author Response · Authors · 2022-08-01
> **Response 4 to Reviewer 9wod**
>
> **Comment 4:** ***"When evaluating the proposed method on several vision backdoor attacks, why only report the evaluation results of altering learning rate and training epochs? What’s the performance of the proposed reparameterized PET on such vision models?'"***
>
> **Response:** PET methods (tuning a few parameters while keeping other parameters frozen) are bound to pre-trained models, and thus can not be applied to non-pre-trained vision models. For vision experiments, we experiment on pre-trained CV models. However, up to now, PET methods are mainly applied to pre-trained language models, and are seldom applied to the pre-trained CV models. Therefore, in this paper, we only report the evaluation results of altering learning rate and training epochs for vision experiments. To address your concerns, we will add the experiments of reparameterized PET on pre-trained vision models in the revised paper.

---

> ### Author Response · Authors · 2022-08-01
> **Response 3 to Reviewer  9wod**
>
> **Comment 3:** ***"The proposed method seems not very effective against several advanced NLP backdoor attacks. As shown in Fig.2-4, Table.1,3. The syntactic, sentence and style-based triggers remain effective after the defense (over 50\% ASR after defense), which might indicate the proposed method is not general and can not handle more stealthy triggers than simple word-level triggers."***
>
> **Response:** (1) Firstly, the high ASR of the backdoor attacks using the so-called "advanced" triggers, e.g., the syntactic trigger, may be attributed to the reason that such triggers **change the semantic information of the text sample dramatically** [1]. It is even possible that the syntactic paraphrase **changes the ground-truth label of texts**. For example, one original sentence from SST-2 is "neither funny nor suspenseful nor particularly well-drawn", which is labeled as negative. After using the syntactic paraphrase in [2], it is transformed into "when it 's funny , it 's nice and tight", which is near positive. Therefore, the model is reasonable to "misclassify" samples with such triggers, which is irrelevant to backdoor attacks.
>
> To further prove this viewpoint, we have performed experiments to see the ASR of the syntactic and word-level triggers on the clean models trained under two settings. We fine-tune the RoBERTa-BASE model on SST-2 with 10 epochs. We also use reparameterized LoRA with the bottleneck dimension 1 to train the model. The experimental results are shown in Table 2. From the experimental results, we can see that even on clean models, the ASR of the syntactic trigger may be above 20\%, which is significantly higher than the word-level trigger. To sum up, the success of syntactic triggers is with the cost of significantly changing or even flipping the semantic information of the samples. Thus, the high ASR value after defense does not indicate that our defense method is ineffective.
>
> |    Trigger Type     | Word-level  ACC | Word-level ASR | Syntactic ACC | Syntactic ASR |
> | :-----------------: | :---------------: | :---------------: | :--------------: | :--------------: |
> | Finetune |       94.23       |       **7.13**        |      94.23       |      **19.96**       |
> |        LoRA         |       92.97       |       **9.76**        |      92.97       |      **20.83**       |
>
> Table 2: The ACC and ASR after training the model with clean training data on SST-2 for word-level triggers and syntactic triggers.
>
> (2) Secondly, even for the advanced NLP backdoor attacks, **our defense method outperforms other backdoor defense methods**, including Backdoor Keyword Identification (BKI) [3], ONION [4], STRIP [5] and RAP [6]. As shown in Table 3, our proposed method achieves **significantly lower ASR** with a very small degradation on the ACC.
>
> | Defender               | ONION  | BKI   | STRIP  | RAP    | Our Method |
> | ---------------------- | ------ | ----- | ------ | ------ | ---------- |
> | Word-level (ACC)     | 92.42  | 94.29 | 94.07  | 94.29  | 94.23      |
> | Word-level (ASR)     | 10.20  | 76.75|  99.12 |  82.89 | **7.89**       |
> | Syntactic (ACC)      |  92.75 | 93.74 |93.85   |  93.52 | 91.98      |
> | Syntactic (ASR)      | 86.29 | 93.09 |  89.47 | 91.67  | **42.11**      |
> | Add-sentence (ACC)   | 93.68  | 94.56 | 94.34  | 93.74  | 92.81      |
> | Add-sentence (ASR)   |99.89  | 100.00 | 100.00 |87.61  | **42.21**      |
> | Style-Transfer (ACC) | 93.47 | 94.18 | 94.07  |  86.00 | 91.76      |
> | Style-Transfer (ASR) | 81.58  |80.48  | 85.09 |  85.53 | **42.21**      |
>
> Table 3: Comparisons of the defense performance between our proposed method and other defense methods against the word-level, syntactic, add-sentence, and style transfer attacks on SST-2.
>
> ----------
>
>
> **References:**
>
> [1] Rethink the evaluation for attack strength of backdoor attacks in natural language processing.
>
> [2] Hidden killer: Invisible textual backdoor attacks with syntactic trigger. In ACL 2021
>
> [3] Mitigating backdoor attacks in lstm-based text classification systems by backdoor keyword identification. Neurocomputing, 2021
>
> [4] Onion: A simple and effective defense against textual backdoor attacks. In EMNLP 2021
>
> [5]  Design and evaluation of a multi-domain trojan detection method on deep neural networks. IEEE Transactions on Dependable and Secure Computing, 2021.
>
> [6] Rap: Robustness-aware perturbations for defending against backdoor attacks on nlp models. In EMNLP 2021

---

> ### Author Response · Authors · 2022-08-01
> **Response 2 to Reviewer 9wod**
>
> **Comment 2.2:** ***"Based on my understanding, the authors assumed the poisoning happens during the fine-tuning stage, i.e. the attackers poisoned the dataset used in the fine-tuning. If so, such a scenario seems too narrow for me.''***
>
> **Response:**  Defending against the poisoning attack in the fine-tuning stage is a critical problem, and training data poisoning is the mainstream backdoor attack in the NLP community. The reasons are as follows:
>
> (1) It has become a **routine** for NLP practitioners to **outsource the curation of training data** to obtain large-scale training datasets [1], and there are many platforms (such as the Huggingface Datasets Library) releasing such training datasets, with millions of downloads (https://huggingface.co/datasets). The released datasets may be poisoned by the attacker and raise serious security concerns. Therefore, building secure machine learning systems against data poisoning attacks is an important problem from the industry perspective [2].
>
> (2) The "Pre-train and then fine-tune'' paradigm has boosted the performance of many downstream AI tasks [3] and become a **mainstream paradigm** for NLP tasks [4]. Thus, defending against the poisoning attack in the fine-tuning stage is a critical problem in the NLP community, which is the main focus of our paper.
>
> Moreover,  our proposed defense methods can be **widely applied to many real-world scenarios**.  As stated in ABL [5], the defense methods under such a setting could benefit companies, research institutes, or government agencies who have the resources to train their own models but rely on outsourced training data. It also benefits MLaaS (Machine Learning as a Service) providers such as Amazon ML and SageMaker, Microsoft Azure AI Platform, Google AI Platform and IBM Watson Machine Learning to help users train backdoor-free models.
>
> ----------
>
> **Comment 2.3:**  ***"What if the attack happens during the pre-training stage? Or users want to train their models from scratch. Under such situations, it’s unknown whether the model’s training still strictly follows moderate-fitting and overfitting stages. If not, please further clarify the threat model in the revision."***
>
> **Response:** The main focus of this paper is the backdoor-free training during fine-tuning a PLM towards downstream tasks, rather than training from scratch. Pre-trained models have been widely used to **boost the performance** of downstream AI tasks [3] and become the **foundation models** for NLP tasks [4]. Although training from scratch is not our focus, we have also performed experiments to see the phenomenon when users train their models from scratch. The results are shown in appendix B.2 in the supplementary material. We perform experiments with a randomly initialized model whose architecture is the same as RoBERTa-BASE on SST-2. From the experiments shown in Table 2 in the appendix, we can see that if users train their models from scratch, the model's training may not follow moderate-fitting and overfitting stages. This demonstrates that pre-training may be an important factor for the defense performance. Following your suggestions, we will further clarify the threat model in the revised paper.
>
> ----------
>
> **References:**
>
> [1] Dataset security for machine learning: Data poisoning, backdoor attacks, and defenses. IEEE Transactions on Pattern Analysis and Machine Intelligence, 2022.
>
> [2] Adversarial machine learning-industry perspectives. In 2020 IEEE Security and Privacy Workshops.
>
> [3] On the opportunities and risks of foundation models. ArXiv, abs/2108.07258, 2021.
>
> [4] Pre-trained models: Past, present and future. ArXiv preprint, abs/2106.07139, 2021.
>
> [5] Anti-backdoor learning: Training clean models on poisoned data. In NIPS 2021.

---

> ### Author Response · Authors · 2022-08-01
> **Response 1 to Reviewer 9wod**
>
> We thank the reviewer for the insightful and constructive feedback for improving this paper. Please find below our point-to-point response to your comments.
>
> **Comment 1.1:**  ***"The proposed method shares similarity with several existing works in vision domains. The overreaching goal of this work is to train a clean model even if the dataset is poisoned. There are several similar works for the vision model backdoor defense[1-2].''***
>
> **Response:**  Thank you for the valuable comment. Overall, our method is different from DBD [1] and ABL [2]. DBD and ABL are both multi-stage methods and they both contain one stage to select potentially poisoned samples. However, our method does not contain multiple stages and we do not need to perform additional operations on the training samples.
>
> ----------
>
> **Comment 1.2:**  ***"Although they were originally proposed for vision models. I would like to see more discussion about the possibility to extend such methods for PLMs and compare them with the proposed method. It will be more convincing if the proposed method outperforms them.''***
>
> **Response:** Thank you for the constructive suggestion. It is possible to extend DBD [1] and ABL [2] to the NLP domain. Following your suggestions, we have adapted DBD and  ABL  to the NLP domain, and experimented with BERT-BASE model under our setting on SST-2 dataset. For the implementation of DBD, we use MixText [3] to replace MixMatch [4]. As shown in Table 1, after applying our defense method, the ASR decreases to **18.86\%**, which is significantly lower compared with DBD (**94.63\%**) and ABL (**99.45\%**). Furthermore, our method has a minor effect on the accuracy of the original task (denoted as ACC). As shown in Table 1, on the BERT-BASE model, after applying our defense method, the ACC is 90.66\%, which is higher than DBD (87.10\%) and ABL (90.12\%). The above results demonstrate that our method significantly **outperforms** both baselines.
>
> The reasons why ABL and DBD can not work well for PLMs may be as follows. ABL is designed for non-pre-trained CNN models and it is not appropriate for pre-trained models. Specifically, after early training, they select examples with the lowest loss values as potentially poisoned samples, which may not be applicable to the PLM. For DBD, only training the classifier to select high-credible samples may be insufficient for the PLM. Also, DBD uses semi-supervised learning in the last stage, which may cause the final accuracy to decline. These may be the reasons why ABL and DBD do not work well for PLMs.
>
> | Defense Method |  DBD  |  ABL  | Our Method |
> | :------------: | :---: | :---: | :--------: |
> |   SST-2 (ACC)   | 87.10 | 90.12 |   90.66    |
> |   SST-2 (ASR)   | 94.63 | 99.45 |   **18.86**    |
>
> Table 1: Comparisons with adapted defense methods from the vision domain.
>
> ----------
>
> **Comment 2.1:** ***"The paper organization should be further improved. For example, the authors did not provide the threat model explicitly to describe the attacker and defender’s capability. ''***
>
> **Response:**  We thank the reviewer for the constructive comment. We clarify our threat model as follows:
>
> (1) The attacker poisons the training data and releases the poisoned training dataset on open-source platforms. The attacker does not control the model training process.
>
> (2) The victim downloads the poisoned training dataset from the open-source platform to train the model. If no defense is applied, the victim will get a model injected with backdoors. However, with our proposed defense method, the victim will get a backdoor-free model even using the poisoned dataset to train the model.
>
> Following your suggestion, we will modify the paper organization and illustrate the threat model more explicitly in the revised paper.
>
> ----------
>
> **References:**
>
> [1] Backdoor defense via decoupling the training process. In ICLR 2021.
>
> [2] Anti-backdoor learning: Training clean models on poisoned data. In NIPS 2021.
>
> [3] MixText: Linguistically-informed interpolation of hidden space for semi-supervised text classification. In ACL 2020.
>
> [4] Mixmatch: A holistic approach to semi-supervised learning. In NIPS 2019.

---

> ### Author Response · Authors · 2022-08-05
> **Looking forward to your reply. (to Reviewer 9wod)**
>
> Dear reviewer,
>
> Thank you for your valuable suggestions and constructive comments. We have done the point-to-point response to your comments. Could you please let us know whether our responses address your concerns? We would really appreciate it if you could let us know before the reviewer-author discussion period ends.

---

> > ### Comment · Reviewer_9wod · 2022-08-08
> > **Response to authors**
> >
> > Thank authors for their comprehensive comments and add-on experimental results, specially for adapting DBD and ABL on NLP backdoor defense. Since most of my concerns are addressed, I will raise my score from 4 to 6.  Beside PICCOLO[1], [2] is another related work which should also be cited in the revision.
> >
> >
> > [1]  Piccolo: Exposing complex backdoors in nlp transformer models. In 2022 IEEE Symposium on Security and Privacy (SP).
> >
> > [2] Constrained Optimization with Dynamic Bound-scaling for Effective NLP Backdoor Defense. Proceedings of the 39th International Conference on Machine Learning (ICML 2022)

---

> > > ### Author Response · Authors · 2022-08-08
> > > **Response to Reviewer 9wod**
> > >
> > > We sincerely thank you for your effort to improve this paper and your decision. DBS [1] is indeed an important related work. Following your suggestion, we will cite the paper [1] in the revision.
> > >
> > > References:
> > >
> > > [1] Constrained Optimization with Dynamic Bound-scaling for Effective NLP Backdoor Defense. Proceedings of the 39th International Conference on Machine Learning (ICML 2022)

---

### Official Review · Reviewer_hehg · 2022-07-18

**Rating:** 7
**Confidence:** 3
**Soundness:** 3 good
**Presentation:** 4 excellent
**Contribution:** 3 good

**Summary:**

In this paper, the authors investigate an interesting phenomenon that when the models are doing a moderate fitting with parameter-efficient training methods, the models are likely to ignore the backdoored features, as those features are ill-trained. Based on this observation, the authors suggest restricting the language model fine-tuning to the moderate-fitting stage to naturally improve the robustness of language models against backdoor triggers. Furthermore, the authors find that (1) parameter capacity, (2) training epochs, and (3) learning rate are key factors that can impact the models’ vulnerability to backdoors. Reducing those hyper-parameters can help models fail to adapt to backdoor features. The authors also make several ablation studies including the visualizations of the training dynamics given different hyper-parameters, the poisoning ratio, and the experiments on the cv models, and draw several interesting conclusions.

**Questions:**

Could you please make it clear about the setup of the backdoor attacks considered in the paper?


**Limitations:**

Please refer to the weakness of the paper.


**Strengths And Weaknesses:**

**Strengths**:
- The paper is well written and easy to follow
- The paper draws several interesting observations and insights into the robustness of parameter-efficient training.
- Though simple, the paper provides an easy yet efficient way to improve model robustness against backdoor attacks.

**Weaknesses**:
- It would be better to illustrate the setting of the backdoor attacks more explicitly. It seems that the backdoor is not injected in the pretraining stage, but injected in the task-specific backdoor fine-tuning stage, which follows the so-called “clean fine-tuning” setting of Qi et al. [1]
- This leads to the second question of whether the proposed method can defend against the backdoors injected in the pre-training stage like BadPre [2] or weight poisoning attack [3]. Such attacks manipulate the backdoored representations dramatically when trigger words appear, which makes me a bit unsure about whether the proposed method will work.

[1] Fanchao Qi, Mukai Li, Yangyi Chen, Zhengyan Zhang, Zhiyuan Liu, Yasheng Wang, and Maosong Sun. Hidden killer: Invisible textual backdoor attacks with syntactic trigger. In ACL 2021
[2] Kangjie Chen, Yuxian Meng, Xiaofei Sun, Shangwei Guo, Tianwei Zhang, Jiwei Li, Chun Fan. BadPre: Task-agnostic Backdoor Attacks to Pre-trained NLP Foundation Models. In ICLR 2022
[3] Keita Kurita, Paul Michel, and Graham Neubig. Weight poisoning attacks on pre-trained models. In ACL 2020

---

> ### Author Response · Authors · 2022-08-01
> **Response to Reviewer hehg**
>
> We thank the reviewer for the insightful and constructive feedback for improving this paper. Please find below our point-to-point response to your comments.
>
> **Comment 1:** ***"It would be better to illustrate the setting of the backdoor attacks more explicitly. It seems that the backdoor is not injected in the pretraining stage, but injected in the task-specific backdoor fine-tuning stage, which follows the so-called “clean fine-tuning” setting of Qi et al. [1]"***
>
> **Response:** Thank you for the constructive suggestion. In our scenario, the attacker poisons the training dataset and releases it on the open-source platform. Then the victim downloads the poisoned training dataset and uses it to fine-tune a PLM (pre-trained language model). Our defense is applied in the fine-tuning stage, enabling the victim to train a backdoor-free model on a poisoned dataset. Following your suggestion, we will modify the paper to illustrate our setting more explicitly in the revised paper.
>
> We also want to point out that there are some differences between our setting and the "clean fine-tuning" setting of Qi et al. [1]. In their setting, the attacker first uses a poisoned dataset of the target task to fine-tune the pre-trained model and obtain a backdoored model. After that, the backdoored model is further fine-tuned using a clean dataset. Differently, under our setting, through the proposed backdoor-free training method, the victim can train a backdoor-free model even using the poisoned dataset. Our defense does not require access to a clean dataset.
>
> ----------
> **Comment 2:** ***"This leads to the second question of whether the proposed method can defend against the backdoors injected in the pre-training stage like BadPre [2] or weight poisoning attack [3]. Such attacks manipulate the backdoored representations dramatically when trigger words appear, which makes me a bit unsure about whether the proposed method will work."***
>
> **Response:** Our defense is not designed for the settings of BadPre [2] and weight poisoning attack [3]. BadPre [2] and weight poisoning attack [3] are model-level attacks. However, our defense is designed for data-level attacks. For model-level attacks like BadPre [2] and weight poisoning attack [3], the attacker can control the model training process to train a backdoored model and release it to open-source platforms like huggingface. However, for the data-level attack, the attacker poisons the training data and releases the poisoned training dataset to open-source platforms. The attacker does not control the model training process. The victim downloads the poisoned training dataset from the open-source platform to train their models. If no defense is applied, the victim will get a backdoored model. However, with our proposed backdoor-free fine-tuning method, the victim will get a backdoor-free model.
>
> Note that data-level attacks are real-world threats to machine learning models. As the training data requirements grow, practitioners have to outsource the curation of training data to obtain large enough training datasets [4] for training their models. In this real-world scenario, the attacker can poison the training dataset but cannot control the model training process. Our proposed defense method is not designed for the setting of BadPre [2] and weight poisoning attack [3]. We expect future works to take inspiration from our findings and design corresponding defense methods for the setting of BadPre [2] and weight poisoning attack [3].
>
> ----------
> **References:**
>
> [1] Fanchao Qi, Mukai Li, Yangyi Chen, Zhengyan Zhang, Zhiyuan Liu, Yasheng Wang, and Maosong Sun. Hidden killer: Invisible textual backdoor attacks with syntactic trigger. In ACL 2021
>
> [2] Kangjie Chen, Yuxian Meng, Xiaofei Sun, Shangwei Guo, Tianwei Zhang, Jiwei Li, Chun Fan. BadPre: Task-agnostic Backdoor Attacks to Pre-trained NLP Foundation Models. In ICLR 2022
>
> [3] Keita Kurita, Paul Michel, and Graham Neubig. Weight poisoning attacks on pre-trained models. In ACL 2020
>
> [4] Micah Goldblum, Dimitris Tsipras, Chulin Xie, Xinyun Chen, Avi Schwarzschild, Dawn Song, Aleksander Madry,
> Bo Li, and Tom Goldstein. Dataset security for machine learning: Data poisoning, backdoor attacks, and defenses. In IEEE Transactions on Pattern Analysis and Machine Intelligence 2022

---

> > ### Comment · Reviewer_hehg · 2022-08-08
> > **Thank you for the response!**
> >
> > I appreciate the authors' response. I am glad to see that my questions are well answered, and the quality of the paper is improved.
> >
> > I maintain my original score.

---

> > > ### Author Response · Authors · 2022-08-08
> > > **Response to Reviewer hehg**
> > >
> > > We sincerely thank you for your effort to improve this paper and your decision.

---

### Author Response · Authors · 2022-08-07
**Summary of Revisions**

We thank the reviewers for their valuable suggestions and constructive comments. Following the reviewers' suggestions, we have revised our manuscript and submitted a new revised version in the "rebuttal revision" field. In the following, we summarize the primary changes we have made for your convenience to check our revised manuscript. The revised parts are highlighted in blue color for easier review.

(1) We have further clarified our threat model in Line 40-45 of the revised manuscript.

(2) We have added the comparisons with other defense methods in Line 279-288 of the revised manuscript.

(3) We have cited PICCOLO [1] and DBS [2] in the related work (Line 94-96) and analyzed the difference between our work with them.

----------

References:

[1] Piccolo: Exposing complex backdoors in nlp transformer models. In 2022 IEEE Symposium on Security and Privacy (SP).

[2] Constrained Optimization with Dynamic Bound-scaling for Effective NLP Backdoor Defense. Proceedings of the 39th International Conference on Machine Learning (ICML 2022)

---

### Meta-Review · Area_Chair_3aUx · 2022-08-25

**Recommendation:** Accept
**Confidence:** Certain

**Metareview:**

The paper proposed an approach to against backdoor triggers by restricting the language model fine-tuning to the moderate-fitting stage. The paper also provides a nice analysis to demonstrate the factors that impact the models’ vulnerability to backdoors. Overall, the paper is well-written and provide sufficient analyses to support the claims. The revision and rebuttal address the comments from the reviewers.

**Award:**

No

---

### Decision · Program_Chairs · 2022-09-14

Accept